# Provable Non-linear Inductive Matrix Completion

**Kai Zhong**
Amazon
kaizhong@amazon.com

**Zhao Song**
University of Washington
magic.linuxkde@gmail.com

**Prateek Jain**
Microsoft
prajain@microsoft.com

**Inderjit S. Dhillon**
Amazon & University of Texas at Austin
isd@amazon.com

## Abstract

Consider a standard recommendation/retrieval problem where given a query, the goal is to retrieve the most relevant items. Inductive matrix completion (IMC) method is a standard approach for this problem where the given query as well as the items are embedded in a common *low-dimensional* space. The inner product between a query embedding and an item embedding reflects relevance of the (query, item) pair. Non-linear IMC (NIMC) uses non-linear networks to embed the query as well as items, and is known to be highly effective for a variety of tasks, such as video recommendations for users, semantic web search, etc. Despite its wide usage, existing literature lacks rigorous understanding of NIMC models. A key challenge in analyzing such models is to deal with the non-convexity arising out of non-linear embeddings in addition to the non-convexity arising out of the low-dimensional restriction of the embedding space, which is akin to the low-rank restriction in the standard matrix completion problem. In this paper, we provide the first theoretical analysis for a simple NIMC model in the realizable setting, where the relevance score of a (query, item) pair is formulated as the inner product between their single-layer neural representations. Our results show that under mild assumptions we can recover the ground truth parameters of the NIMC model using standard (stochastic) gradient descent methods if the methods are initialized within a small distance to the optimal parameters. We show that a standard tensor method can be used to initialize the solution within the required distance to the optimal parameters. Furthermore, we show that the number of query-item relevance observations required, a key parameter in learning such models, scales nearly linearly with the input dimensionality thus matching existing results for the standard *linear* inductive matrix completion.

## 1   Introduction

Real-world recommendation systems and information retrieval systems aim to obtain the relevance between "queries" and "items", such as user-item ratings, query-web relevance, query-product relevance, etc. A classic technique to model a recommendation system is matrix completion or collaborative filtering [CR09, GUH16] where the model is learned from a few observed user-item ratings without the need for the entire user/item information. Modern recommendation systems also have access to a large amount of side information about the users and the items. In the meantime, the development of deep learning models has facilitated the extraction of effective neural representations for the users/items. Therefore modern recommendation systems such as Youtube video recommendation [CAS16], image recommendation [LLL$^+$16], music recommendation [WW14], etc, are adopting deep learning representations for users/items. On the other hand, modern information re-

| Dataset | #movies | #users | # ratings | # movie feat. | # user feat. | NIMC | IMC |
|---------|---------|--------|-----------|---------------|--------------|-------|-------|
| ml-100k | 1682 | 943 | 100,000 | 39 | 29 | **1.034** | 1.321 |
| ml-1m | 3883 | 6040 | 1,000,000 | 38 | 29 | **1.021** | 1.320 |

**Table 1:** Test RMSE for recommending new users with existing movies on Movielens dataset. We use users' demographic information and movies' genre information as features $x$ and $y$ respectively. We randomly split the users into existing users (training data) and new users (testing data) with ratio 4:1. Hence, we are predicting ratings for completely new users, for which only users' demographic features are available. The user features include 21 types of occupations, 7 different age ranges and one gender information; the movie features include 18-19 (18 for ml-1m and 19 for ml-100k) genre features and 20 features from the top 20 right singular values of the training rating matrix (which has size #training users -by- #movies). $k$ is set to 50, and $\phi$ is the ReLU activation.

trieval systems [HHG$^+$13, NSM$^+$19] are also evolving from lexical relevance between queries and documents to deep semantic relevance by leveraging deep learning techniques.

A common practice for modeling the relevance for query-item pair is to first extract representations from the original features of the query/item. For instance, convolutional neural networks are used to represent videos/images and recurrent neural networks/attention-based models are used to extract the embeddings for text features. Then the representations of the pair are used to calculate the relevance by some similarity function, such as cosine similarity or inner product.

In particular, if we use inner product, the query($x$)-item($y$) relevance can be modeled as follows.

$$A(x, y) = \langle \mathcal{U}(x), \mathcal{V}(y) \rangle, \tag{1}$$

where $x \in \mathbb{R}^{d_1}, y \in \mathbb{R}^{d_2}$ are the feature vectors, $A(x, y)$ is their relevance and $\mathcal{U} : \mathbb{R}^{d_1} \to \mathbb{R}^k, \mathcal{V} : \mathbb{R}^{d_2} \to \mathbb{R}^k$ are non-linear mappings from the raw feature space to the latent space. We call this model non-linear inductive matrix completion (NIMC) model named after the linear inductive matrix completion model [JD13].

Despite the practical success of deep learning models in real-world recommendation and information retrieval systems, a rigorous understanding for why they work is still lacking. Theoretical analyses for the linear case of NIMC [ABEV06], where $\mathcal{U}, \mathcal{V}$ are linear mappings, have been provided in [JD13, XJZ13, CHD15], where the challenges mainly come from the non-convexity arising from the restriction to the low-dimensional embedding space. For non-linear cases, an additional major challenge is the non-convexity from non-linear feature extraction mappings.

In this paper, we provide analyses for a simple one-layer neural network style NIMC model. That is, we set $\mathcal{U}(x) = \phi(U^{*\top}x)$ and $\mathcal{V}(x) = \phi(V^{*\top}x)$ where $\phi$ is a nonlinear activation function $\phi$ and $U \in \mathbb{R}^{d_1 \times k}, V \in \mathbb{R}^{d_2 \times k}$ ($k \leq d_1, k \leq d_2$). Despite the seemingly simple non-linearity of one-layer neural network, we can still see there is significant improvement over linear IMC, for example on the Movielens [Gro97] dataset as shown in Table 1.

Note that if $\phi$ is ReLU then the latent space is guaranteed to be in non-negative orthant which in itself can be a desirable property for certain recommendation problems.

In particular, we formulate a squared-loss based optimization problem for estimating parameters $U^*$ and $V^*$. We show that under a realizable model and Gaussian input assumption, the objective function is locally strongly convex within a "reasonably large" neighborhood of the ground truth. Moreover, we show that the above strong convexity claim holds even if the number of observed relevance scores is nearly-linear in dimension and polynomial in the conditioning of the weight matrices. In particular, for well-conditioned matrices, we can recover the underlying parameters using only $\mathrm{poly}\log(d_1 + d_2)$ query-item relevance scores, which is critical for practical recommendation systems as they tend to have very few relevance scores available per query. Our analysis covers popular activation functions, e.g., sigmoid and ReLU, and unearths various subtleties that arise due to the activation function. Finally we discuss how we can leverage standard tensor decomposition techniques to initialize our parameters well. We would like to stress that practitioners typically use random initialization itself, and hence results studying random initialization for NIMC model would be of significant interest.

As mentioned above, due to non-linearity of activation function along with non-convexity of the parameter space, the existing proof techniques do not apply directly to the problem. Moreover, we

have to carefully argue about both the optimization landscape as well as the sample complexity of the algorithm which is not carefully studied for neural networks. Our proof establishes some new techniques that might be of independent interest, e.g., how to handle the redundancy in the parameters for ReLU activation. To the best of our knowledge, this is one of the first theoretically rigorous study of neural-network based recommendation/retrieval systems and will hopefully be a stepping stone for similar analysis for "deeper" neural networks based recommendation systems. We would also like to highlight that our model can be viewed as a strict generalization of a one-hidden layer neural network, hence our result represents one of the few rigorous guarantees for models that are more powerful than one-hidden layer neural networks [LY17, BGMSS18, ZSJ$^+$17]. Finally, we apply our model on synthetic datasets and verify our theoretical analysis.

In summary, the main contribution of this paper is to provide, as far as we know, the first theoretical recovery guarantees for learning a neural network based inductive matrix completion model using gradient descent when the parameters are initialized by tensor methods.

## 1.1 Related work

*Collaborative filtering*: Our model is a non-linear version of the standard inductive matrix completion model [JD13]. Practically, IMC has been applied to gene-disease prediction [ND14], matrix sensing [ZJD15], multi-label classification[YJKD14], blog recommender system [SCLD15], link prediction [CHD15] and semi-supervised clustering [CHD15, SCH$^+$16]. However, IMC restricts the latent space of users/items to be a linear transformation of the user/item's feature space. [SCH$^+$16] extended the model to a three-layer neural network and showed significantly better empirical performance for multi-label/multi-class classification problem and semi-supervised problems.

Although standard IMC has linear mappings, it is still a non-convex problem due to the bilinearity $UV^\top$. To deal with this non-convex problem, [JD13, Har14] provided recovery guarantees using alternating minimization with sample complexity linear in dimension. [XJZ13] relaxed this problem to a nuclear-norm problem and also provided recovery guarantees. More general norms have been studied [RSW16, SWZ17a, SWZ17b, SWZ18], e.g. weighted Frobenius norm, entry-wise $\ell_1$ norm. More recently, [ZDG18] uses gradient-based non-convex optimization and proves a better sample complexity. [CHD15] studied dirtyIMC models and showed that the sample complexity can be improved if the features are informative when compared to matrix completion. Several low-rank matrix sensing problems [ZJD15, GJZ17] are also closely related to IMC models where the observations are sampled only from the diagonal elements of the relevance matrix. [Ren10, LY16] introduced and studied an alternate framework for relevance prediction with side-information but the prediction function is linear in their case as well.

*Neural networks*: Nonlinear activation functions play an important role in neural networks. Recently, several powerful results have been discovered for learning one-hidden-layer feedforward neural networks [Tia17, ZSJ$^+$17, JSA15, LY17, BGMSS18, GKKT17, VW18, ZYWG18], convolutional neural networks [BG17, ZSD17, DLT18a, DLT$^+$18b, GKM18, DWZ$^+$18]. However, our problem is a strict generalization of the one-hidden layer neural network and is not covered by the above mentioned results.

**Notation.** For any function $f$, we define $\widetilde{O}(f)$ to be $f \cdot \log^{O(1)}(f)$. For two functions $f, g$, we use the shorthand $f \lesssim g$ (resp. $\gtrsim$) to indicate that $f \leq Cg$ (resp. $\geq$) for an absolute constant $C$. We use $f \eqsim g$ to mean $cf \leq g \leq Cf$ for constants $c, C$. We use $\mathrm{poly}(f)$ to denote $f^{O(1)}$.

**Roadmap.** We first present the formal model and the corresponding optimization problem in Section 2. We then present the local strong convexity and local linear convergence results in Section 3. Finally, we show simulation results to verify our theory (Section 4).

## 2 Problem Formulation

Consider a query-item recommender/retrieval system, where we have $n_1$ queries with feature vectors $X := \{x_i\}_{i \in [n_1]} \subseteq \mathbb{R}^{d_1}$, $n_2$ items with feature vectors $Y := \{y_j\}_{j \in [n_2]} \subseteq \mathbb{R}^{d_2}$ and a collection of partially-observed query-item relevance scores, $\mathcal{A}_{\mathrm{obs}} = \{A(x, y) | (x, y) \in \Omega \subseteq X \times Y\}$. That is $A(x_i, y_j)$ is the relevance score that query $x_i$ gave for item $y_j$. For simplicity, we assume $x_i$'s and $y_j$'s are sampled i.i.d. from distribution $\mathcal{X}$ and $\mathcal{Y}$, respectively. Each element of the index set $\Omega$ is also sampled independently and uniformly with replacement from $S := X \times Y$.

In this paper, our goal is to predict the relevance score for *any* query-item pair with feature vectors $x$ and $y$, respectively. We model the query-item relevance scores as:

$$A(x,y) = \phi(U^{*\top}x)^\top \phi(V^{*\top}y), \tag{2}$$

where $U^* \in \mathbb{R}^{d_1 \times k}$, $V^* \in \mathbb{R}^{d_2 \times k}$ and $\phi$ is a *non-linear* activation function. Under this realizable model, our goal is to *recover* $U^*, V^*$ from a collection of observed entries, $\{A(x,y)|(x,y) \in \Omega\}$. Without loss of generality, we set $d_1 = d_2$. Also we treat $k$ as a constant throughout the paper. Our analysis requires $U^*, V^*$ to be full column rank, so we require $k \leq d$. And w.l.o.g., we assume $\sigma_k(U^*) = \sigma_k(V^*) = 1$, i.e., the smallest singular value of both $U^*$ and $V^*$ is 1.

Note that this model is similar to one-hidden layer feed-forward network popular in standard classification/regression tasks. However, as there is an inner product between the output of two non-linear layers, $\phi(U^*x)$ and $\phi(V^*y)$, it cannot be modeled by a single hidden layer neural network (with same number of nodes). Also, for linear activation function, the problem reduces to inductive matrix completion [ABEV06, JD13].

Now, to solve for $U^*, V^*$, we optimize a simple squared-loss based optimization problem, i.e.,

$$\min_{U \in \mathbb{R}^{d_1 \times k}, V \in \mathbb{R}^{d_2 \times k}} f_\Omega(U, V),$$

where

$$f_\Omega(U, V) = \sum_{(x,y) \in \Omega} (\phi(U^\top x)^\top \phi(V^\top y) - A(x,y))^2. \tag{3}$$

Naturally, the above problem is a challenging non-convex optimization problem that is strictly harder than two non-convex optimization problems which are challenging in their own right: a) the linear inductive matrix completion where non-convexity arises due to bilinearity of $U^\top V$, and b) the standard one-hidden layer neural network (NN). In fact, recently a lot of research has focused on understanding various properties of both the linear inductive matrix completion problem [GJZ17, JD13] as well as one-hidden layer NN [GLM18, ZSJ$^+$17].

In this paper, we show that despite the non-convexity of Problem (3), it behaves as a convex optimization problem close to the optima if the data is sampled stochastically from a Gaussian distribution. This result combined with standard tensor decomposition based initialization [ZSJ$^+$17, KCL15, JSA15] leads to a polynomial time algorithm for solving (3) optimally if the data satisfies certain sampling assumptions in Theorem 2.1. Moreover, we also discuss the effect of various activation functions, especially the difference between a sigmoid activation function vs RELU activation (see Theorem 3.2 and Theorem 3.4).

Informally, our recovery guarantee can be stated as follows,

**Theorem 2.1** (Informal Recovery Guarantee)**.** *Consider a recommender system with a realizable model Eq. (2) with sigmoid activation, Assume the features $\{x_i\}_{i \in [n_1]}$ and $\{y_j\}_{j \in [n_2]}$ are sampled i.i.d. from the normal distribution and the observed pairs $\Omega$ are i.i.d. sampled from $\{x_i\}_{i \in [n_1]} \times \{y_j\}_{j \in [n_2]}$ uniformly at random. Then there exists an algorithm such that $U^*, V^*$ can be recovered to any precision $\epsilon$ with time complexity and sample complexity (refers to $n_1, n_2, |\Omega|$) polynomial in the dimension and the condition number of $U^*, V^*$, and logarithmic in $1/\epsilon$.*

## 3   Main Results

Our main result shows that when initialized properly, gradient-based algorithms will be guaranteed to converge to the ground truth. We first study the Hessian of empirical risk for different activation functions, then based on the positive-definiteness of the Hessian for smooth activations, we show local linear convergence of gradient descent. The proof sketch is provided in Appendix C.

The positive definiteness of the Hessian *does not* hold for several activation functions. Here we provide some examples.

**Counter Example 1)** The Hessian at the ground truth for linear activation is not positive definite because for any full-rank matrix $R \in \mathbb{R}^{k \times k}$, $(U^*R, V^*R^{-1})$ is also a global optimal.

**Counter Example 2)** The Hessian at the ground truth for ReLU activation is not positive definite because for any diagonal matrix $D \in \mathbb{R}^{k \times k}$ with positive diagonal elements, $U^*D, V^*D^{-1}$ is also

a global optimal. These counter examples have a common property: there is redundancy in the parameters. Surprisingly, for sigmoid and tanh, the Hessian around the ground truth is positive definite. More surprisingly, we will later show that for ReLU, if the parameter space is constrained properly, its Hessian at a given point near the ground truth can also be proved to be positive definite with high probability.

## 3.1 Local Geometry and Local Linear Convergence for Sigmoid and Tanh

We define two natural condition numbers for the problem that captures the "hardness" of the problem:

**Definition 3.1.** *Define $\lambda := \max\{\lambda(U^*), \lambda(V^*)\}$ and $\kappa := \max\{\kappa(U^*), \kappa(V^*)\}$, where $\lambda(U) = \sigma_1^k(U)/(\Pi_{i=1}^k \sigma_i(U))$, $\kappa(U) = \sigma_1(U)/\sigma_k(U)$, and $\sigma_i(U)$ denotes the $i$-th singular value of $U$ with the ordering $\sigma_i \geq \sigma_{i+1}$.*

First we show the result for sigmoid and tanh activations.

**Theorem 3.2** (Positive Definiteness of Hessian for Sigmoid and Tanh). *Let the activation function $\phi$ in the NIMC model (2) be sigmoid or tanh and let $\kappa, \lambda$ be as defined in Definition 3.1. Then for any $t > 1$ and any given $U, V$, if*

$$n_1 \gtrsim t\lambda^4\kappa^2 d\log^2 d, \quad n_2 \gtrsim t\lambda^4\kappa^2 d\log^2 d,$$
$$|\Omega| \gtrsim t\lambda^4\kappa^2 d\log^2 d,$$
$$\|U - U^*\| + \|V - V^*\| \lesssim 1/(\lambda^2\kappa),$$

*then with probability at least $1 - d^{-t}$, the smallest eigenvalue of the Hessian of Eq. (3) is lower bounded by:*

$$\lambda_{\min}(\nabla^2 f_\Omega(U, V)) \gtrsim 1/(\lambda^2\kappa).$$

**Remark.** Theorem 3.2 shows that, given sufficiently large number of query-item relevance scores and a sufficiently large number of queries/items themselves, the Hessian at a point close enough to the true parameters $U^*$, $V^*$, is positive definite with high probability. The sample complexity, including $n_1, n_2$ and $|\Omega|$, have a near-linear dependency on the dimension, which matches the linear IMC analysis [JD13]. Strong convexity parameter as well as the sample complexity depend on the condition number of $U^*, V^*$ as defined in Definition 3.1. Although we don't explicitly show the dependence on $k$, both sample complexity and the minimal eigenvalue scale as a polynomial of $k$. The proofs can be found in Appendix C.

As the above theorem shows the Hessian is positive definite w.h.p. for a given $U, V$ that is close to the optima. This result along with smoothness of the activation function implies linear convergence of gradient descent that samples a fresh batch of samples in each iteration as shown in the following, whose proof is postponed to Appendix E.1.

**Theorem 3.3.** *Let $[U^c, V^c]$ be the parameters in the $c$-th iteration. Assuming $\|U^c - U^*\| + \|V^c - V^*\| \lesssim 1/(\lambda^2\kappa)$, then given a fresh sample set, $\Omega$, that is independent of $[U^c, V^c]$ and satisfies the conditions in Theorem 3.2, the next iterate using one step of gradient descent, i.e., $[U^{c+1}, V^{c+1}] = [U^c, V^c] - \eta\nabla f_\Omega(U^c, V^c)$, satisfies*

$$\|U^{c+1} - U^*\|_F^2 + \|V^{c+1} - V^*\|_F^2$$
$$\leq (1 - M_l/M_u)(\|U^c - U^*\|_F^2 + \|V^c - V^*\|_F^2)$$

*with probability $1 - d^{-t}$, where $\eta = \Theta(1/M_u)$ is the step size and $M_l \gtrsim 1/(\lambda^2\kappa)$ is the lower bound on the eigenvalues of the Hessian and $M_u \lesssim 1$ is the upper bound on the eigenvalues of the Hessian.*

**Remark.** The linear convergence requires each iteration has a set of fresh samples. However, since it converges linearly to the ground-truth, we only need $\log(1/\epsilon)$ iterations, therefore the sample complexity is only logarithmic in $1/\epsilon$. This dependency is better than directly using Tensor decomposition method [JSA15], which requires $O(1/\epsilon^2)$ samples. Note that we only use Tensor decomposition to initialize the parameters. Therefore the sample complexity required in our tensor initialization does not depend on $\epsilon$.

## 3.2 Empirical Hessian around the Ground Truth for ReLU

We now present our result for ReLU activation. As we see in Counter Example 2, without any further modification, the Hessian for ReLU is not locally strongly convex due to the redundancy in parameters. Therefore, we reduce the parameter space by fixing one parameter for each $(u_i, v_i)$ pair, $i \in [k]$. In particular, we fix $u_{1,i} = u_{1,i}^*, \forall i \in [k]$ when minimizing the objective function, Eq. (3), where $u_{1,i}$ is $i$-th element in the first row of $U$. Note that as long as $u_{1,i}^* \neq 0$, $u_{1,i}$ can be fixed to any other non-zero values. We set $u_{1,i} = u_{1,i}^*$ just for simplicity of the proof. The new objective function can be represented as

$$f_\Omega^{\text{ReLU}}(W, V) = \frac{1}{2|\Omega|} \times \sum_{(x,y)\in\Omega} (\phi(W^\top x_{2:d} + x_1(u^{*(1)})^\top)^\top \phi(V^\top y) - A(x,y))^2. \tag{4}$$

where $u^{*(1)}$ is the first row of $U^*$ and $W \in \mathbb{R}^{(d-1)\times k}$.

Surprisingly, after fixing one parameter for each $(u_i, v_i)$ pair, the Hessian using ReLU is also positive definite w.h.p. for a given $(U, V)$ around the ground truth.

**Theorem 3.4** (Positive Definiteness of Hessian for ReLU). *Define* $u_0 := \min_{i\in[k]}\{|u_{1,i}^*|\}$. *For any* $t > 1$ *and any given* $U, V$, *if*

$$n_1 \gtrsim u_0^{-4} t\lambda^4 \kappa^{12} d \log^2 d, \quad n_2 \gtrsim u_0^{-4} t\lambda^4 \kappa^{12} d \log^2 d,$$
$$|\Omega| \gtrsim u_0^{-4} t\lambda^4 \kappa^{12} d \log^2 d,$$
$$\|W - W^*\| + \|V - V^*\| \lesssim u_0^4/\lambda^4 \kappa^{12},$$

*then with probability* $1 - d^{-t}$, *the minimal eigenvalue of the objective for ReLU activation function, Eq. (4), is lower bounded,*

$$\lambda_{\min}(\nabla^2 f_\Omega^{\text{ReLU}}(W, V)) \gtrsim u_0^2/\lambda^2 \kappa^4.$$

**Remark.** Similar to the sigmoid/tanh case, the sample complexity for ReLU case also has a linear dependency on the dimension. However, here we have a worse dependency on the condition number of the weight matrices. The scale of $u_0$ can also be important and in practice one needs to set it carefully. Note that although the activation function is not smooth, the Hessian at a given point can still exist with probability 1, since ReLU is smooth almost everywhere and there are only a finite number of samples. However, owing to the non-smoothness, a proof of convergence of gradient descent method for ReLU is still an open problem.

## 3.3 Technical Analysis Challenges

At high level the proofs for Theorem 3.2 and Theorem 3.4 include the following steps. 1) Show that the population Hessian at the ground truth is positive definite. 2) Show that population Hessians near the ground truth are also positive definite. 3) Employ matrix Bernstein inequality to bound the population Hessian and the empirical Hessian.

Here we show the challenges to prove the positive definiteness of the Hessian for the population risk at the ground truth. The population risk for Eq. (3) is given by:

$$f_\mathcal{D}(U, V) = \frac{1}{2} \underset{(x,y)\sim\mathcal{D}}{\mathbb{E}}[(\phi(U^\top x)^\top \phi(V^\top y) - A(x,y))^2], \tag{5}$$

where $\mathcal{D} := \mathcal{X} \times \mathcal{Y}$.

Let the Hessian of $f_\mathcal{D}(U, V)$ at the ground-truth $(U, V) = (U^*, V^*)$ be $H^* \in \mathbb{R}^{(2dk)\times(2dk)}$, which can be decomposed into the following two types of blocks ($i \in [k], j \in [k]$),

$$\frac{\partial^2 f_\mathcal{D}(U^*, V^*)}{\partial u_i \partial u_j} = \underset{x,y}{\mathbb{E}}\left[\phi'(u_i^{*\top}x)\phi'(u_j^{*\top}x)xx^\top \phi(v_i^{*\top}y)\phi(v_j^{*\top}y)\right],$$
$$\frac{\partial^2 f_\mathcal{D}(U^*, V^*)}{\partial u_i \partial v_j} = \underset{x,y}{\mathbb{E}}\left[\phi'(u_i^{*\top}x)\phi'(v_j^{*\top}y)xy^\top \phi(v_i^{*\top}y)\phi(u_j^{*\top}x)\right].$$

To study the positive definiteness of $H^*$, we characterize the minimal eigenvalue of $H^*$ by a constrained optimization problem,

$$\lambda_{\min}(H^*) = \min_{(a,b) \in \mathbb{B}} \mathbb{E}_{x,y} \left[ \left( \sum_{i=1}^{k} \phi'(u_i^{*\top} x) \phi(v_i^{*\top} y) x^\top a_i + \phi'(v_i^{*\top} y) \phi(u_i^{*\top} x) y^\top b_i \right)^2 \right], \quad (6)$$

where $(a,b) \in \mathbb{B}$ denotes that $\sum_{i=1}^{k} \|a_i\|^2 + \|b_i\|^2 = 1$. Obviously, $\lambda_{\min}(H^*) \geq 0$ due to the squared loss and the realizable assumption. However, this is not sufficient for the local convexity around the ground truth, which requires the positive (semi-)definiteness for the *neighborhood* around the ground truth. In other words, we need to show that $\lambda_{\min}(H^*)$ is strictly greater than 0, so that we can characterize an area in which the Hessian still preserves positive definiteness (PD) despite the deviation from the ground truth.

As we mentioned previously there are activation functions that lead to redundancy in parameters. Hence one challenge is to distill properties of the activation functions that preserve the PD. Another challenge is the correlation introduced by $U^*$ when it is non-orthogonal. So we first study the minimal eigenvalue for orthogonal $U^*$ and orthogonal $V^*$ and then link the non-orthogonal case to the orthogonal case.

### 3.4 Initialization

To achieve the ground truth, our algorithm needs a good initialization method that can initialize the parameters to fall into the neighborhood of the ground truth. Here we show that this is possible by using tensor method under the Gaussian assumption. In the following, we consider estimating $U^*$. Estimating $V^*$ is similar.

Define a third-order moment of the input,

$$M_3 := \mathbb{E}[A(x,y) \cdot (x^{\otimes 3} - x \widetilde{\otimes} I)],$$

where $x \widetilde{\otimes} I := \sum_{j=1}^{d} [x \otimes e_j \otimes e_j + e_j \otimes x \otimes e_j + e_j \otimes e_j \otimes x]$. Define $\gamma_j(\sigma) := \mathbb{E}[\phi(\sigma \cdot z) z^j], \ \forall j = 0, 1, 2, 3$. Then under the Gaussian input assumption, $M_3 = \sum_{i=1}^{k} \alpha_i \overline{u}_i^{*\otimes 3}$, where $\overline{u}_i^* = u_i^*/\|u_i^*\|, \alpha_i = \gamma_0(\|v_i^*\|) (\gamma_3(\|u_i^*\|) - 3\gamma_1(\|u_i^*\|))$. When $\alpha_i \neq 0$, we can approximately recover $\alpha_i$ and $\overline{u}_i^*$ from the empirical version of $M_3$ using non-orthogonal tensor decomposition [KCL15].

Although tensor initialization has nice theoretical guarantees and sample complexity, it heavily depends on Gaussian assumption and realizable model assumption. In contrast, practitioners typically use random initialization.

## 4 Simulation

In this section, we generate some synthetic datasets to verify the sample complexity and the convergence of gradient descent. For simplicity, we apply gradient descent with initialization as $W^{(0)} = (1 - \alpha)W^* + \alpha W^{(r)}$, where $W^*$ is the ground truth, $W^{(r)}$ is a Gaussian random matrix, and $\alpha \in [0, 1]$. In Fig. 1 (a)(b), $\alpha = 0.1$ and in Fig. 1 (c)(d) $\alpha = 1$. The sampling rule for the observations follows our previous assumptions. For each $n, m$ pair, we make 5 trials and take the average of the successful recovery times. We say a solution $(U, V)$ successfully recovers the ground truth parameters when the solution achieves 0.001 relative testing error, i.e.,

$$\|\phi(X_t U)\phi(X_t U)^\top - \phi(X_t U^*)\phi(X_t U^*)^\top\|_F \leq 0.001 \cdot \|\phi(X_t U^*)\phi(X_t U^*)^\top\|_F,$$

where $X_t \in \mathbb{R}^{n \times d}$ is a newly sampled testing dataset. For both ReLU and sigmoid, we minimize the original objective function (3).

In (a), $k = 5, d = 10, n = 1000$, and $m = 10000$. As we can see, (a) shows how the objective value converges, which is almost linear. In (b), we show how initialization affects the recovery rate. When $k, d$ are large ($k = 10, d = 100, n = 500$) and the initialization is purely random ($\alpha = 1$), gradient descent doesn't converge to the ground truth. As shown in Fig. 1 (c)(d), when $k = 5, d = 10$, pure random initialization can converge to the ground truth. We believe that it is because when $k, d$ are

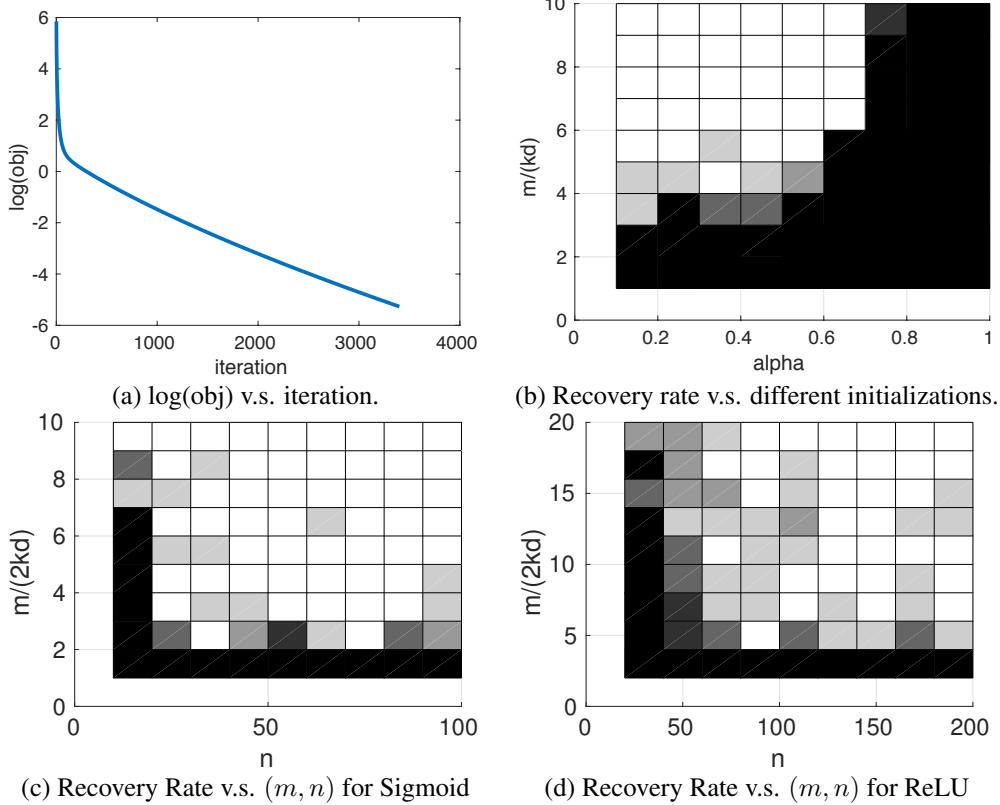

(a) log(obj) v.s. iteration.

(b) Recovery rate v.s. different initializations.

(c) Recovery Rate v.s. $(m, n)$ for Sigmoid

(d) Recovery Rate v.s. $(m, n)$ for ReLU

**Figure 1:** (a) shows how the gradient descent converges, which is almost linear. (b) shows that when $k, d$ are large ($k = 10, d = 100$) and the initialization is purely random ($\alpha = 1$), gradient descent doesn't converge to the ground truth. (c) and (d) show the recovery rate for sigmoid and ReLU activation functions respectively. White blocks denote 100% recovery rate over 5 trials, while black means 100% failure.

larger, random initialization can be further away from the ground truth. Hence, gradient descent can get stuck in local optima more easily.

We illustrate the recovery rate and sample complexity for sigmoid and ReLU in Figure 1 (c)(d). For sigmoid, set the number of samples $n_1 = n_2 = n = \{10 \cdot i\}_{i=1,2\cdots,10}$ and the number of observations $|\Omega| = m = \{2kd \cdot i\}_{i=1,2,\cdots,10}$. For ReLU, set $n = \{20 \cdot i\}_{i=1,2\cdots,10}$ and $m = \{4kd \cdot i\}_{i=1,2,\cdots,10}$.

As we can see, ReLU requires more samples/observations than that for sigmoid for exact recovery (note the scales of $n$ and $m/2kd$ are different in the two figures). This is consistent with our theoretical results. Comparing Theorem 3.2 and Theorem 3.4, we can see the sample complexity for ReLU has a worse dependency on the conditioning of $U^*, V^*$ than sigmoid. We can also see that when $n$ is sufficiently large, the number of observed ratings required remains the same for both methods. This is also consistent with the theorems, where $|\Omega|$ is near-linear in $d$ and is independent of $n$.

## 5 Conclusions

In this paper, we propose a nonlinear IMC model that represents one of the simplest inductive models for neural-network-based recommendation/retrieval systems. We study local geometry of the empirical risk function and show that, close to the optima, the function is strongly convex for both ReLU and sigmoid activations. Therefore, using a smooth activation function like sigmoid activation, gradient descent recovers the underlying model with polynomial sample complexity and time complexity if the parameters are initialized by standard tensor methods. Thus we provide the first theoretically rigorous result for the non-linear recommendation/retrieval system problem, which we hope will spur further progress on the theory of deep-learning based recommendation/retrieval systems.

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
