[Supplementary Material]

# Appendix

## A  Notation

For any positive integer $n$, we use $[n]$ to denote the set $\{1, 2, \cdots, n\}$. For random variable $X$, let $\mathbb{E}[X]$ denote the expectation of $X$ (if this quantity exists).

For any vector $x \in \mathbb{R}^n$, we use $\|x\|$ to denote its $\ell_2$ norm.

We provide several definitions related to matrix $A$. Let $\det(A)$ denote the determinant of a square matrix $A$. Let $A^\top$ denote the transpose of $A$. Let $A^\dagger$ denote the Moore-Penrose pseudoinverse of $A$. Let $A^{-1}$ denote the inverse of a full rank square matrix. Let $\|A\|_F$ denote the Frobenius norm of matrix $A$. Let $\|A\|$ denote the spectral norm of matrix $A$. Let $\sigma_i(A)$ to denote the $i$-th largest singular value of $A$.

We use $\mathbf{1}_f$ to denote the indicator function, which is 1 if $f$ holds and 0 otherwise. Let $I_d \in \mathbb{R}^{d \times d}$ denote the identity matrix. We use $\phi(z)$ to denote an activation function. We use $\mathcal{D}$ to denote a Gaussian distribution $\mathcal{N}(0, I_d)$. For integer $k$, we use $\mathcal{D}_k$ to denote $\mathcal{N}(0, I_k)$.

For any function $f$, we define $\widetilde{O}(f)$ to be $f \cdot \log^{O(1)}(f)$. In addition to $O(\cdot)$ notation, for two functions $f, g$, we use the shorthand $f \lesssim g$ (resp. $\gtrsim$) to indicate that $f \leq Cg$ (resp. $\geq$) for an absolute constant $C$. We use $f \approx g$ to mean $cf \leq g \leq Cf$ for constants $c, C$.

## B  Preliminaries

We state some useful facts in this section.

**Fact B.1.** *Let* $A = [a_1 \quad a_2 \quad \cdots \quad a_k]$. *Let* $\mathrm{diag}(A) \in \mathbb{R}^k$ *denote the vector where the $i$-th entry is* $A_{i,i}, \forall i \in [k]$. *Let* $\mathbf{1} \in \mathbb{R}^k$ *denote the vector that the $i$-th entry is* 1, $\forall i \in [k]$. *We have the following properties,*

$$\text{(I)} \quad \sum_{i=1}^{k}(a_i^\top e_i)^2 = \| \mathrm{diag}(A)\|_2^2,$$

$$\text{(II)} \quad \sum_{i=1}^{k}(a_i^\top a_i)^2 = \|A\|_F^2,$$

$$\text{(III)} \quad \sum_{i=1}^{k}\sum_{j=1}^{k}(a_i^\top a_j) = \|A \cdot \mathbf{1}\|_2^2,$$

$$\text{(IV)} \quad \sum_{i \neq j} a_i^\top a_j = \|A \cdot \mathbf{1}\|_2^2 - \|A\|_F^2.$$

*Proof.* Using the definition, it is easy to see that (I), (II) and (III) are holding.

Proof of (IV), we have

$$\sum_{i \neq j} a_i^\top a_j = \sum_{i,j} a_i^\top a_j - \sum_{i=1}^{k} a_i^\top a_i = \|A \cdot \mathbf{1}\|_2^2 - \|A\|_F^2.$$

where the last step follows by (II) and (III). $\qquad\square$

**Fact B.2.** *Let* $A = [a_1 \quad a_2 \quad \cdots \quad a_k]$. *Let* $\mathrm{diag}(A) \in \mathbb{R}^k$ *denote the vector where the $i$-th entry is* $A_{i,i}, \forall i \in [k]$. *Let* $\mathbf{1} \in \mathbb{R}^k$ *denote the vector that the $i$-th entry is* 1, $\forall i \in [k]$. *We have the following*

*properties,*

$$\text{(I)} \sum_{i \neq j} a_i^\top e_i e_i^\top a_j = (\text{diag}(A)^\top \cdot (A \cdot \mathbf{1})) - \|\text{diag}(A)\|_2^2,$$

$$\text{(II)} \sum_{i \neq j} a_i^\top e_j e_j^\top a_j = (\text{diag}(A)^\top \cdot (A \cdot \mathbf{1})) - \|\text{diag}(A)\|_2^2,$$

$$\text{(III)} \sum_{i \neq j} a_i^\top e_i a_j^\top e_j = (\text{diag}(A)^\top \cdot \mathbf{1})^2 - \|\text{diag}(A)\|_2^2,$$

$$\text{(IV)} \sum_{i \neq j} a_i^\top e_j a_j^\top e_i = \langle A^\top, A \rangle - \|\text{diag}(A)\|_2^2.$$

*Proof.* Proof of (I). We have

$$\begin{aligned}
\sum_{i \neq j} a_i^\top e_i e_i^\top a_j &= \sum_{i,j} a_i^\top e_i e_i^\top a_j - \sum_{i=1}^k a_i^\top e_i e_i^\top a_i \\
&= \sum_{i,j} a_{i,i} e_i^\top a_j - \|\text{diag}(A)\|_2^2 \\
&= \sum_{i=1}^k a_{i,i} e_i^\top \sum_{j=1}^k a_j - \|\text{diag}(A)\|_2^2 \\
&= (\text{diag}(A)^\top \cdot (A \cdot \mathbf{1})) - \|\text{diag}(A)\|_2^2
\end{aligned}$$

Proof of (II). It is similar to (I).

Proof of (III). We have

$$\begin{aligned}
\sum_{i \neq j} a_i^\top e_i a_j^\top e_j &= \sum_{i,j} a_i^\top e_i a_j^\top e_j - \sum_{i=1}^k a_i^\top e_i a_i^\top e_i \\
&= \sum_{i=1}^k a_i^\top e_i \cdot \sum_{j=1}^k a_j^\top e_j - \sum_{i=1}^k a_i^\top e_i a_i^\top e_i \\
&= \sum_{i=1}^k a_{i,i} \cdot \sum_{j=1}^k a_{j,j} - \sum_{i=1}^k a_{i,i} a_{i,i} \\
&= (\text{diag}(A)^\top \cdot \mathbf{1})^2 - \|\text{diag}(A)\|_2^2
\end{aligned}$$

Proof of (IV). We have

$$\begin{aligned}
\sum_{i \neq j} a_i^\top e_j a_j^\top e_i &= \sum_{i \neq j} \text{tr}[a_i^\top e_j a_j^\top e_i] \\
&= \sum_{i \neq j} \text{tr}[e_j a_j^\top e_i a_i^\top] \\
&= \sum_{i \neq j} \langle e_j a_j^\top, a_i e_i^\top \rangle \\
&= \sum_{i,j} \langle e_j a_j^\top, a_i e_i^\top \rangle - \sum_{i=1}^k \langle e_i a_i^\top, a_i e_i^\top \rangle \\
&= \langle A^\top, A \rangle - \|\text{diag}(A)\|_2^2.
\end{aligned}$$

where the second step follows by $\text{tr}[ABCD] = \text{tr}[BCDA]$, the third step follows by $\text{tr}[AB] = \langle A, B^\top \rangle$. $\qquad\square$

## C   Proof Sketch

At high level the proofs for Theorem 3.2 and Theorem 3.4 include the following steps. 1) Show that the population Hessian at the ground truth is positive definite. 2) Show that population Hessians near the ground truth are also positive definite. 3) Employ matrix Bernstein inequality to bound the population Hessian and the empirical Hessian.

We now formulate the Hessian. The Hessian of Eq. (3), $\nabla^2 f_\Omega(U, V) \in \mathbb{R}^{(2kd) \times (2kd)}$, can be decomposed into two types of blocks, ($i \in [k], j \in [k]$),

$$\frac{\partial^2 f_\Omega(U, V)}{\partial u_i \partial v_j}, \frac{\partial^2 f_\Omega(U, V)}{\partial u_i \partial u_j},$$

where $u_i (v_j$, resp.) is the $i$-th column of $U$ ($j$-th column of $V$, resp.). Note that each of the above second-order derivatives is a $d \times d$ matrix.

The first type of blocks are given by:

$$\frac{\partial^2 f_\Omega(U, V)}{\partial u_i \partial v_j} = \widehat{\mathbb{E}}_\Omega \left[ \phi'(u_i^\top x) \phi'(v_j^\top y) x y^\top \phi(v_i^\top y) \phi(u_j^\top x) \right] + \delta_{ij} \widehat{\mathbb{E}}_\Omega \left[ h_{x,y}(U, V) \phi'(u_i^\top x) \phi'(v_i^\top y) x y^\top \right],$$

where $\widehat{\mathbb{E}}_\Omega[\cdot] = \frac{1}{|\Omega|} \sum_{(x,y) \in \Omega} [\cdot]$, $\delta_{ij} = 1_{i=j}$, and

$$h_{x,y}(U, V) = \phi(U^\top x)^\top \phi(V^\top y) - \phi(U^{*\top} x)^\top \phi(V^{*\top} y).$$

For sigmoid/tanh activation function, the second type of blocks are given by:

$$\frac{\partial^2 f_\Omega(U, V)}{\partial u_i \partial u_j} = \widehat{\mathbb{E}}_\Omega \left[ \phi'(u_i^\top x) \phi'(u_j^\top x) x x^\top \phi(v_i^\top y) \phi(v_j^\top y) \right] + \delta_{ij} \widehat{\mathbb{E}}_\Omega \left[ h_{x,y}(U, V) \phi''(u_i^\top x) \phi(v_i^\top y) x x^\top \right]. \tag{7}$$

For ReLU/leaky ReLU activation function, the second type of blocks are given by:

$$\frac{\partial^2 f_\Omega(U, V)}{\partial u_i \partial u_j} = \widehat{\mathbb{E}}_\Omega \left[ \phi'(u_i^\top x) \phi'(u_j^\top x) x x^\top \phi(v_i^\top y) \phi(v_j^\top y) \right].$$

Note that the second term of Eq. (7) is missing here as $(U, V)$ are fixed, the number of samples is finite and $\phi''(z) = 0$ almost everywhere.

In this section, we will discuss important lemmas/theorems for Step 1 in Appendix C.1 and Step 2,3 in Appendix C.3.

### C.1   Positive definiteness of the population hessian

The corresponding population risk for Eq. (3) is given by:

$$f_\mathcal{D}(U, V) = \frac{1}{2} \mathbb{E}_{(x,y) \sim \mathcal{D}} [(\phi(U^\top x)^\top \phi(V^\top y) - A(x, y))^2], \tag{8}$$

where $\mathcal{D} := \mathcal{X} \times \mathcal{Y}$. For simplicity, we also assume $\mathcal{X}$ and $\mathcal{Y}$ are normal distributions.

Now we study the Hessian of the population risk at the ground truth. Let the Hessian of $f_\mathcal{D}(U, V)$ at the ground-truth $(U, V) = (U^*, V^*)$ be $H^* \in \mathbb{R}^{(2dk) \times (2dk)}$, which can be decomposed into the following two types of blocks ($i \in [k], j \in [k]$),

$$\frac{\partial^2 f_\mathcal{D}(U^*, V^*)}{\partial u_i \partial u_j} = \mathbb{E}_{x,y} \left[ \phi'(u_i^{*\top} x) \phi'(u_j^{*\top} x) x x^\top \phi(v_i^{*\top} y) \phi(v_j^{*\top} y) \right],$$

$$\frac{\partial^2 f_\mathcal{D}(U^*, V^*)}{\partial u_i \partial v_j} = \mathbb{E}_{x,y} \left[ \phi'(u_i^{*\top} x) \phi'(v_j^{*\top} y) x y^\top \phi(v_i^{*\top} y) \phi(u_j^{*\top} x) \right].$$

To study the positive definiteness of $H^*$, we characterize the minimal eigenvalue of $H^*$ by a constrained optimization problem,

$$\lambda_{\min}(H^*) = \min_{(a,b) \in \mathbb{B}} \mathbb{E}_{x,y} \left[ \left( \sum_{i=1}^k \phi'(u_i^{*\top} x) \phi(v_i^{*\top} y) x^\top a_i + \phi'(v_i^{*\top} y) \phi(u_i^{*\top} x) y^\top b_i \right)^2 \right], \tag{9}$$

where $(a, b) \in \mathbb{B}$ denotes that $\sum_{i=1}^{k} \|a_i\|^2 + \|b_i\|^2 = 1$. Obviously, $\lambda_{\min}(H^*) \geq 0$ due to the squared loss and the realizable assumption. However, this is not sufficient for the local convexity around the ground truth, which requires the positive (semi-)definiteness for the *neighborhood* around the ground truth. In other words, we need to show that $\lambda_{\min}(H^*)$ is strictly greater than 0, so that we can characterize an area in which the Hessian still preserves positive definiteness (PD) despite the deviation from the ground truth.

**Challenges.** As we mentioned previously there are activation functions that lead to redundancy in parameters. Hence one challenge is to distill properties of the activation functions that preserve the PD. Another challenge is the correlation introduced by $U^*$ when it is non-orthogonal. So we first study the minimal eigenvalue for orthogonal $U^*$ and orthogonal $V^*$ and then link the non-orthogonal case to the orthogonal case.

## C.2 Warm up: orthogonal case

In this section, we consider the case when $U^*, V^*$ are unitary matrices, i.e., $U^{*\top}U^* = U^*U^{*\top} = I_d$. $(d = k)$. This case is easier to analyze because the dependency between different elements of $x$ or $y$ can be disentangled. And we are able to provide lower bound for the Hessian. Before we introduce the lower bound, let's first define the following quantities for an activation function $\phi$.

$$
\begin{aligned}
\alpha_{i,j} &:= \mathop{\mathbb{E}}_{z \sim \mathcal{N}(0,1)} [(\phi(z))^i z^j], \\
\beta_{i,j} &:= \mathop{\mathbb{E}}_{z \sim \mathcal{N}(0,1)} [(\phi'(z))^i z^j], \\
\gamma &:= \mathop{\mathbb{E}}_{z \sim \mathcal{N}(0,1)} [\phi(z)\phi'(z)z], \\
\rho &:= \min\{(\alpha_{2,0}\beta_{2,0} - \alpha_{1,0}^2\beta_{1,0}^2 - \beta_{1,0}^2\alpha_{1,1}^2), \ (\alpha_{2,0}\beta_{2,2} - \alpha_{1,0}^2\beta_{1,2}^2 - \gamma^2)\}.
\end{aligned}
\tag{10}
$$

We now present a lower bound for general activation functions including sigmoid and tanh.

**Lemma C.1.** *Let $(a, b) \in \mathbb{B}$ denote that $\sum_{i=1}^{k} \|a_i\|^2 + \|b_i\|^2 = 1$. Assume $d = k$ and $U^*, V^*$ are unitary matrices, i.e., $U^{*\top}U^* = U^*U^{*\top} = V^*V^{*\top} = V^{*\top}V^* = I_d$, then the minimal eigenvalue of the population Hessian in Eq. (9) can be simplified as,*

$$
\min_{(a,b) \in \mathbb{B}} \mathop{\mathbb{E}}_{x,y} \left[ \left( \sum_{i=1}^{k} \phi'(x_i)\phi(y_i)x^\top a_i + \phi'(y_i)\phi(x_i)y^\top b_i \right)^2 \right].
$$

*Let $\beta, \rho$ be defined as in Eq. (10). If the activation function $\phi$ satisfies $\beta_{1,1} = 0$, then $\lambda_{\min}(H^*) \geq \rho$.*

Since sigmoid and tanh have symmetric derivatives w.r.t. 0, they satisfy $\beta_{1,1} = 0$. Specifically, we have $\rho \approx 0.000658$ for sigmoid and $\rho \approx 0.0095$ for tanh. Also for ReLU, $\beta_{1,1} = 1/2$, so ReLU does not fit in this lemma. The full proof of Lemma C.1, the lower bound of the population Hessian for ReLU and the extension to non-orthogonal cases can be found in Appendix D.

## C.3 Error bound for the empirical Hessian near the ground truth

In the previous section, we have shown PD for the population Hessian at the ground truth for the orthogonal cases. Based on that, we can characterize the landscape around the ground truth for the empirical risk. In particular, we bound the difference between the empirical Hessian near the ground truth and the population Hessian at the ground truth. The theorem below provides the error bound w.r.t. the number of samples $(n1, n2)$ and the number of observations $|\Omega|$ for both sigmoid and ReLU activation functions.

**Theorem C.2.** *For any $\epsilon > 0$, if*

$$
n_1 \gtrsim \epsilon^{-2}td\log^2 d, \ n_2 \gtrsim \epsilon^{-2}td\log^2 d, \ |\Omega| \gtrsim \epsilon^{-2}td\log^2 d,
$$

*then with probability at least $1 - d^{-t}$, for sigmoid/tanh,*

$$
\|\nabla^2 f_\Omega(U, V) - \nabla^2 f_\mathcal{D}(U^*, V^*)\| \lesssim \epsilon + \|U - U^*\| + \|V - V^*\|;
$$

*for ReLU,*

$$
\|\nabla^2 f_\Omega(U, V) - \nabla^2 f_\mathcal{D}(U^*, V^*)\| \lesssim \left( \|V - V^*\|^{1/2} + \|U - U^*\|^{1/2} + \epsilon \right) (\|U^*\| + \|V^*\|)^2.
$$

The key idea to prove this theorem is to use the population Hessian at $(U, V)$ as a bridge.

On one side, we bound the population Hessian at the ground truth and the population Hessian at $(U, V)$. This would be easy if the second derivative of the activation function is Lipschitz, which is the case of sigmoid and tanh. But ReLU doesn't have this property. However, we can utilize the condition that the parameters are close enough to the ground truth and the piece-wise linearity of ReLU to bound this term.

On the other side, we bound the empirical Hessian and the population Hessian. A natural idea is to apply matrix Bernstein inequality. However, there are two obstacles. First the Gaussian variables are not uniformly bounded. Therefore, we instead use Lemma B.7 in [ZSJ+17], which is a loosely-bounded version of matrix Bernstein inequality. The second obstacle is that each individual Hessian calculated from one observation $(x, y)$ is not independent from another observation $(x', y')$, since they may share the same feature $x$ or $y$. The analyses for vanilla IMC and MC assume all the items(users) are given and the observed entries are independently sampled from the whole matrix. However, our observations are sampled from the joint distribution of $\mathcal{X}$ and $\mathcal{Y}$.

To handle the dependency, our model assumes the following two-stage sampling rule. First, the items/users are sampled from their distributions independently, then given the items and users, the observations $\Omega$ are sampled uniformly with replacement. The key question here is how to combine the error bounds from these two stages. Fortunately, we found special structures in the blocks of Hessian which enables us to separate $x, y$ for each block, and bound the errors in stage separately. See Appendix E for details.

## D  Positive Definiteness of Population Hessian

### D.1  Orthogonal case

We first study the orthogonal case, where $d = k$ and $U^*, V^*$ are unitary matrices, i.e., $U^{*\top}U^* = U^*U^{*\top} = V^*V^{*\top} = V^{*\top}V^* = I_d$.

#### D.1.1  Lower bound on minimum eigenvalue

**Lemma D.1** (Restatement of Lemma C.1). *Let $(a, b) \in \mathbb{B}$ denote that $\sum_{i=1}^{k} \|a_i\|^2 + \|b_i\|^2 = 1$. Assume $d = k$ and $U^*, V^*$ are unitary matrices, i.e., $U^{*\top}U^* = U^*U^{*\top} = V^*V^{*\top} = V^{*\top}V^* = I_d$, then the minimal eigenvalue of the population Hessian in Eq. (9) can be simplified as,*

$$\lambda_{\min}(H^*) = \min_{(a,b)\in\mathbb{B}} \mathbb{E}_{x,y} \left[ \left( \sum_{i=1}^{k} \phi'(x_i)\phi(y_i)x^\top a_i + \phi'(y_i)\phi(x_i)y^\top b_i \right)^2 \right]. \tag{11}$$

*Let $\beta, \rho$ be defined as in Eq. (10). If the activation function $\phi$ satisfies $\beta_{1,1} = 0$, then $\lambda_{\min}(H^*) \geq \rho$.*

*Proof.* In the orthogonal case, we can easily transform Eq. (9) to Eq. (11) since $x, y$ are normal distribution. Now we can decompose Eq. (11) into the following three terms.

$$\mathbb{E}_{x,y} \left[ \left( \sum_{i=1}^{k} \phi'(x_i)\phi(y_i)x^\top a_i + \phi'(y_i)\phi(x_i)y^\top b_i \right)^2 \right]$$

$$= \underbrace{\mathbb{E}_{x,y} \left[ \left( \sum_{i=1}^{k} \phi'(x_i)\phi(y_i)x^\top a_i \right)^2 \right] + \mathbb{E}_{x,y} \left[ \left( \sum_{i=1}^{k} \phi'(y_i)\phi(x_i)y^\top b_i \right)^2 \right]}_{C}$$

$$+ \underbrace{2 \mathbb{E}_{x,y} \left[ \sum_{i,j} \phi'(x_i)\phi(y_i)x^\top a_i \phi'(y_j)\phi(x_j)y^\top b_j \right]}_{D}.$$

Note that the first term is similar to the second term, so we just lower bound the first term and the third term. Define $A = [a_1, a_2, \cdots, a_k], B = [b_1, b_2, \cdots, b_k]$. Let $A_o$ be the off-diagonal part of $A$

and $A_d$ be the diagonal part of $A$, i.e., $A_o + A_d = A$. And let $g_A = \operatorname{diag}(A)$ be the vector of the diagonal elements of A. We will bound $C$ and $D$ in the following.

For $C$, we have

$$
\mathbb{E}_{x,y}\left[\left(\sum_{i=1}^{k}\phi'(x_i)\phi(y_i)x^\top a_i\right)^2\right]
$$

$$
= \sum_{i=1}^{k}\mathbb{E}_{x,y}\left[\left(\phi'(x_i)\phi(y_i)x^\top a_i\right)^2\right] + \sum_{i\neq j}\mathbb{E}_{x,y}\left[\phi'(x_i)\phi(y_i)x^\top a_i \cdot \phi'(x_j)\phi(y_j)x^\top a_j\right]
$$

$$
= \sum_{i=1}^{k}\alpha_{2,0}\left[(a_i^\top e_i)^2(\beta_{2,2} - \beta_{2,0}) + \beta_{2,0}\|a_i\|^2\right]
$$

$$
+ \sum_{i\neq j}\alpha_{1,0}^2\left[\beta_{1,0}^2 a_i^\top a_j + (\beta_{1,2}\beta_{1,0} - \beta_{1,0}^2)(a_i^\top e_i e_i^\top a_j + a_i^\top e_j a_j^\top e_j) + \beta_{1,1}^2(a_i^\top e_i a_j^\top e_j + a_i^\top e_j a_j^\top e_i)\right]
$$

$$
= C_1 + C_2.
$$

where the last step follows by

$$
C_1 = \sum_{i=1}^{k}\alpha_{2,0}\left[(a_i^\top e_i)^2(\beta_{2,2} - \beta_{2,0}) + \beta_{2,0}\|a_i\|^2\right]
$$

$$
C_2 = \sum_{i\neq j}\alpha_{1,0}^2\left[\beta_{1,0}^2 a_i^\top a_j + (\beta_{1,2}\beta_{1,0} - \beta_{1,0}^2)(a_i^\top e_i e_i^\top a_j + a_i^\top e_j a_j^\top e_j) + \beta_{1,1}^2(a_i^\top e_i a_j^\top e_j + a_i^\top e_j a_j^\top e_i)\right]
$$

First we can simplify $C_1$ in the following sense,

$$
C_1 = \alpha_{2,0}(\beta_{2,2} - \beta_{2,0})\sum_{i=1}^{k}(a_i^\top e_i)^2 + \alpha_{2,0}\beta_{2,0}\sum_{i=1}^{k}\|a_i\|_2^2
$$

$$
= \alpha_{2,0}(\beta_{2,2} - \beta_{2,0})\|\operatorname{diag}(A)\|_2^2 + \alpha_{2,0}\beta_{2,0}\|A\|_F^2,
$$

where the last step follows by Fact B.1.

We can rewrite $C_2$ in the following sense

$$
C_2 = \alpha_{1,0}^2(\beta_{1,0}^2 C_{2,1} + (\beta_{1,2}\beta_{1,0} - \beta_{1,0}^2) \cdot (C_{2,2} + C_{2,3}) + \beta_{1,1}^2(C_{2,4} + C_{2,5})).
$$

where

$$
C_{2,1} = \sum_{i\neq j}a_i^\top a_j
$$

$$
C_{2,2} = \sum_{i\neq j}a_i^\top e_i e_i^\top a_j
$$

$$
C_{2,3} = \sum_{i\neq j}a_i^\top e_j e_j^\top a_j
$$

$$
C_{2,4} = \sum_{i\neq j}a_i^\top e_i a_j^\top e_j
$$

$$
C_{2,5} = \sum_{i\neq j}a_i^\top e_j a_j^\top e_i
$$

Using Fact B.1, we have

$$
C_{2,1} = \|A \cdot \mathbf{1}\|_2^2 - \|A\|_F^2.
$$

Using Fact B.2, we have

$$
C_{2,2} = (\operatorname{diag}(A)^\top \cdot (A \cdot \mathbf{1})) - \|\operatorname{diag}(A)\|_2^2,
$$

$$
C_{2,3} = (\operatorname{diag}(A)^\top \cdot (A \cdot \mathbf{1})) - \|\operatorname{diag}(A)\|_2^2,
$$

$$
C_{2,4} = (\operatorname{diag}(A)^\top \cdot \mathbf{1})^2 - \|\operatorname{diag}(A)\|_2^2,
$$

$$
C_{2,5} = \langle A^\top, A\rangle - \|\operatorname{diag}(A)\|_2^2.
$$

Thus,

$$
\begin{aligned}
C_2 = \alpha_{1,0}^2(&\beta_{1,0}^2(\|A \cdot \mathbf{1}\|_2^2 - \|A\|_F^2) \\
&+ (\beta_{1,2}\beta_{1,0} - \beta_{1,0}^2)2 \cdot (\operatorname{diag}(A)^\top \cdot (A \cdot \mathbf{1}) - \|\operatorname{diag}(A)\|_2^2) \\
&+ \beta_{1,1}^2((\operatorname{diag}(A)^\top \cdot \mathbf{1})^2 + \langle A^\top, A \rangle - 2\|\operatorname{diag}(A)\|_2^2)).
\end{aligned}
$$

We consider $C_1 + C_2$ by focusing different terms, for the $\|A\|_F^2$(from $C_1$ and $C_2$), we have

$$
(\alpha_{2,0}\beta_{2,0} - \alpha_{1,0}^2\beta_{1,0}^2)\|A\|_F^2.
$$

For the term $\langle A, A^\top \rangle$ (from $C_{2,5}$), we have

$$
\alpha_{1,0}^2\beta_{1,1}^2\langle A, A^\top \rangle.
$$

For the term $\|\operatorname{diag}(A)\|_2^2$ (from $C_1$ and $C_2$), we have

$$
(\alpha_{2,0}(\beta_{2,2} - \beta_{2,0}) - 2\alpha_{1,0}^2(\beta_{1,2}\beta_{1,0} - \beta_{1,0}^2) - 2\alpha_{1,0}\beta_{1,1}^2)\|\operatorname{diag}(A)\|_2^2
$$

For the term $\|A \cdot \mathbf{1}\|_2^2$ (from $C_{2,1}$), we have

$$
\alpha_{1,0}^2\beta_{1,0}^2\|A \cdot \mathbf{1}\|_2^2.
$$

For the term $\operatorname{diag}(A)^\top \cdot A \cdot \mathbf{1}$ (from $C_{2,2}$ and $C_{2,3}$), we have

$$
2\alpha_{1,0}^2(\beta_{1,2}\beta_{1,0} - \beta_{1,0}^2)\operatorname{diag}(A)^\top \cdot A \cdot \mathbf{1}.
$$

For the term $(\operatorname{diag}(A)^\top \cdot \mathbf{1})^2$ (from $C_{2,4}$), we have

$$
\alpha_{1,0}^2\beta_{1,1}^2(\operatorname{diag}(A)^\top \cdot \mathbf{1})^2.
$$

Putting it all together, we have

$$
\begin{aligned}
C_1 + C_2 &= (\alpha_{2,0}\beta_{2,0} - \alpha_{1,0}^2\beta_{1,0}^2)\|A\|_F^2 + \alpha_{1,0}^2\beta_{1,1}^2\langle A, A^\top \rangle \\
&\quad + (\alpha_{2,0}(\beta_{2,2} - \beta_{2,0}) - 2\alpha_{1,0}^2(\beta_{1,2}\beta_{1,0} - \beta_{1,0}^2) - 2\alpha_{1,0}^2\beta_{1,1}^2) \cdot \|\operatorname{diag}(A)\|^2 \\
&\quad + \alpha_{1,0}^2\beta_{1,0}^2\|A \cdot \mathbf{1}\|^2 + 2\alpha_{1,0}^2(\beta_{1,2}\beta_{1,0} - \beta_{1,0}^2)(\operatorname{diag}(A)^\top \cdot A \cdot \mathbf{1}) + \alpha_{1,0}^2\beta_{1,1}^2(\operatorname{diag}(A)^\top \cdot \mathbf{1})^2 \\
&= (\alpha_{2,0}\beta_{2,0} - \alpha_{1,0}^2\beta_{1,0}^2)(\|A_o\|_F^2 + \|g_A\|^2) + \alpha_{1,0}^2\beta_{1,1}^2(\langle A_o, A_o^\top \rangle + \|g_A\|^2) \\
&\quad + (\alpha_{2,0}\beta_{2,2} - \alpha_{2,0}\beta_{2,0} - 2\alpha_{1,0}^2\beta_{1,2}\beta_{1,0} + 2\alpha_{1,0}^2\beta_{1,0}^2 - 2\alpha_{1,0}^2\beta_{1,1}^2) \cdot \|g_A\|^2 \\
&\quad + \alpha_{1,0}^2\beta_{1,0}^2(\|g_A\|^2 + \|A_o \cdot \mathbf{1}\|^2 + 2g_A^\top \cdot A_o \cdot \mathbf{1}) \\
&\quad + 2\alpha_{1,0}^2(\beta_{1,2}\beta_{1,0} - \beta_{1,0}^2)(g_A^\top \cdot A_o \cdot \mathbf{1} + \|g_A\|^2) + \alpha_{1,0}^2\beta_{1,1}^2(g_A^\top \cdot \mathbf{1})^2 \\
&= (\alpha_{2,0}\beta_{2,0} - \alpha_{1,0}^2\beta_{1,0}^2)\|A_o\|_F^2 + \alpha_{1,0}^2\beta_{1,1}^2\langle A_o, A_o^\top \rangle + (\alpha_{2,0}\beta_{2,2} - \alpha_{1,0}^2\beta_{1,1}^2) \cdot \|g_A\|^2 \\
&\quad + \alpha_{1,0}^2\beta_{1,0}^2(\|A_o \cdot \mathbf{1}\|^2) + 2\alpha_{1,0}^2\beta_{1,2}\beta_{1,0}(g_A^\top \cdot A_o \cdot \mathbf{1}) + \alpha_{1,0}^2\beta_{1,1}^2(g_A^\top \cdot \mathbf{1})^2.
\end{aligned}
$$

By doing a series of equivalent transformations, we have removed the expectation and the formula $C$ becomes a form of $A$ and the moments of $\phi$. These equivalent transforms are mainly based on the fact that $x_i, x_j, y_i, y_j$ for any $i \neq j$ are independent on each other.

Similarly we can reformulate $D$,

$$
\begin{aligned}
&\mathbb{E}_{x,y}\left[\sum_{i,j} \phi'(x_i)\phi(y_i)x^\top a_i\phi'(y_j)\phi(x_j)y^\top b_j\right] \\
&= \sum_i \mathbb{E}_{x,y}\left[\phi'(x_i)\phi(y_i)x^\top a_i\phi'(y_ji)\phi(x_i)y^\top b_i\right] + \sum_{i \neq j} \mathbb{E}_{x,y}\left[\phi'(x_i)\phi(y_i)x^\top a_i\phi'(y_j)\phi(x_j)y^\top b_j\right] \\
&= \sum_i \gamma^2 a_i^\top e_i b_i^\top e_i + \sum_{i \neq j} \alpha_{1,1}^2 a_i^\top e_j b_j^\top e_i + \alpha_{1,1}\beta_{1,1}(a_i^\top e_j b_j^\top e_j + a_i^\top e_i b_j^\top e_i) + \beta_{1,1}^2 a_i^\top e_i b_j^\top e_j \\
&= (\gamma^2 - \beta_{1,0}^2\alpha_{1,1}^2 - 2\alpha_{1,0}\alpha_{1,1}\beta_{1,0}\beta_{1,1} - \alpha_{1,0}^2\beta_{1,1}^2)g_A^\top g_B \\
&\quad + \beta_{1,0}^2\alpha_{1,1}^2\langle A, B^\top \rangle + \alpha_{1,0}^2\beta_{1,1}^2(g_A^\top 1)(g_B^\top 1) \\
&\quad + \alpha_{1,0}\alpha_{1,1}\beta_{1,0}\beta_{1,1}[(A1)^\top g_B + (B1)^\top g_A] \\
&= (\gamma^2 - \alpha_{1,0}^2\beta_{1,1}^2)g_A^\top g_B + \beta_{1,0}^2\alpha_{1,1}^2\langle A_o, B_o^\top \rangle + \alpha_{1,0}^2\beta_{1,1}^2(g_A^\top 1)(g_B^\top 1) \\
&\quad + \alpha_{1,0}\alpha_{1,1}\beta_{1,0}\beta_{1,1}[(A_o 1)^\top g_B + (B_o 1)^\top g_A].
\end{aligned}
$$

Combining the above results, we have

$$
\begin{aligned}
\lambda_{\min}(H^*) = \min_{\|A\|_F^2 + \|B\|_F^2 = 1} \Big( &\beta_{1,0}^2 \alpha_{1,1}^2 \|A_o + B_o^\top\|_F^2 \\
&+ \|\alpha_{1,0}\beta_{1,0}A_o 1 + \alpha_{1,0}\beta_{1,2}g_A + \alpha_{1,1}\beta_{1,1}g_B\|^2 \\
&+ \|\alpha_{1,0}\beta_{1,0}B_o 1 + \alpha_{1,0}\beta_{1,2}g_B + \alpha_{1,1}\beta_{1,1}g_A\|^2 \\
&+ (\alpha_{2,0}\beta_{2,0} - \alpha_{1,0}^2\beta_{1,0}^2 - \beta_{1,0}^2\alpha_{1,1}^2 - \alpha_{1,0}^2\beta_{1,1}^2)(\|A_o\|_F^2 + \|B_o\|_F^2) \\
&+ 1/2 \cdot \alpha_{1,0}^2\beta_{1,1}^2 (\|A_o + A_o^\top\|_F^2 + \|B_o + B_o^\top\|_F^2) \\
&+ [\alpha_{2,0}\beta_{2,2} - \alpha_{1,0}^2\beta_{1,1}^2 - \alpha_{1,0}^2\beta_{1,2}^2 - \alpha_{1,1}^2\beta_{1,1}^2] \cdot (\|g_A\|^2 + \|g_B\|^2) \\
&+ 2(\gamma^2 - \alpha_{1,0}^2\beta_{1,1}^2 - 2\alpha_{1,0}\alpha_{1,1}\beta_{1,1}\beta_{1,2})g_A^\top g_B \\
&+ \alpha_{1,0}^2\beta_{1,1}^2(g_A^\top 1 + g_B^\top 1)^2 \Big).
\end{aligned}
\tag{12}
$$

The final output of the above formula has a clear form: most non-negative terms are extracted. $A, B$ are separated into the off-diagonal elements and off-diagonal elements and these two terms can be dealt with independently. Now we consider the activation functions that satisfy $\beta_{1,1} = 0$, which further simplifies the equation. Note that Sigmoid and $\tanh$ satisfy this condition.

Finally, for $\beta_{1,1} = 0$, we obtain

$$
\begin{aligned}
\lambda_{\min}(H^*) &= \min_{\sum_{i=1}^k \|a_i\|^2 + \|b_i\|^2 = 1} \mathbb{E}_{x,y} \left[ \left( \sum_{i=1}^k \phi'(x_i)\phi(y_i)x^\top a_i + \phi'(y_i)\phi(x_i)y^\top b_i \right)^2 \right] \\
&= \min_{\|A\|_F^2 + \|B\|_F^2 = 1} (\alpha_{2,0}\beta_{2,0} - \alpha_{1,0}^2\beta_{1,0}^2 - \beta_{1,0}^2\alpha_{1,1}^2)(\|A_o\|_F^2 + \|B_o\|_F^2) \\
&\quad + (\alpha_{2,0}\beta_{2,2} - \alpha_{1,0}^2\beta_{1,2}^2 - \gamma^2)(\|g_A\|^2 + \|g_B\|^2) \\
&\quad + \beta_{1,0}^2\alpha_{1,1}^2 \|A_o + B_o^\top\|_F^2 + \gamma^2\|g_A + g_B\|^2 \\
&\quad + \alpha_{1,0}^2(\|\beta_{1,0}g_A + \beta_{1,2}A_o 1\|^2 + \alpha_{1,0}^2\|\beta_{1,0}g_A + \beta_{1,2}B_o 1\|^2) \\
&\geq \underbrace{\min\{(\alpha_{2,0}\beta_{2,0} - \alpha_{1,0}^2\beta_{1,0}^2 - \beta_{1,0}^2\alpha_{1,1}^2), (\alpha_{2,0}\beta_{2,2} - \alpha_{1,0}^2\beta_{1,2}^2 - \gamma^2)\}}_{:=\rho}.
\end{aligned}
$$

For sigmoid, we have $\rho = 0.000658$; for tanh, we have $\rho = 0.0095$.

$\square$

The following lemma will be used when transforming non-orthogonal cases to orthogonal cases.

**Lemma D.2.** *For any $A = [a_1, a_2, \cdots, a_k] \in \mathbb{R}^{d \times k}$, we have,*

$$
\mathbb{E}_{x,y \sim \mathcal{D}_k} \left[ \left\| \sum_{i=1}^k \phi'(x_i)\phi(y_i)a_i \right\|^2 \right] \geq (\alpha_{2,0}\beta_{2,0} - \alpha_{1,0}^2\beta_{1,0}^2)\|A\|_F^2.
$$

*Proof.* Recall $1 \in \mathbb{R}^d$ denote the all ones vector.

$$
\begin{aligned}
&\mathbb{E}_{x,y \sim \mathcal{D}_k} \left[ \left\| \sum_{i=1}^k \phi'(x_i)\phi(y_i)a_i \right\|^2 \right] \\
&= \mathbb{E}_{x,y \sim \mathcal{D}_k} \left[ \sum_{i=1}^k (\phi'(x_i)\phi(y_i))^2 \|a_i\|^2 \right] + \mathbb{E}_{x,y \sim \mathcal{D}_k} \left[ \sum_{i \neq j} \phi'(x_i)\phi(y_i)\phi'(x_j)\phi(y_j)a_i^\top a_j \right] \\
&= (\alpha_{2,0}\beta_{2,0} - \alpha_{1,0}^2\beta_{1,0}^2)\|A\|_F^2 + \alpha_{1,0}^2\beta_{1,0}^2\|A \cdot 1\|^2 \\
&\geq (\alpha_{2,0}\beta_{2,0} - \alpha_{1,0}^2\beta_{1,0}^2)\|A\|_F^2.
\end{aligned}
$$

Thus, we complete the proof. $\square$

Now let's show the PD of the population Hessian of Eq. (4) for the ReLU case. where $u^{*(1)}$ is the first row of $U^*$ and $W \in \mathbb{R}^{(d-1) \times k}$.

**Lemma D.3.** *Consider the activation function to be ReLU. Assume $k = d$, $U^*, V^*$ are unitary matrices and $u_{1,i}^* \neq 0, \forall i \in [k]$. Then the minimal eigenvalue of the corresponding population Hessian of Eq. (4) is lower bounded,*

$$\lambda_{\min}(\nabla^2 f_{\mathcal{D}}^{\mathrm{ReLU}}(W^*, V^*)) \gtrsim \min_{i \in [k]} \{u_{1,i}^{*2}\},$$

*where $W^* = U_{2:d,:}^*$ is the last $d-1$ rows of $U^*$ and*

$$f_{\mathcal{D}}^{\mathrm{ReLU}}(W, V) := \mathbb{E}_{x,y} \left[ (\phi(W^\top x_{2:d} + x_1 (u^{*(1)})^\top)^\top \phi(V^\top y) - A(x, y))^2 \right], \tag{13}$$

*Proof.* By fixing $u_{i,1} = u_{i,1}^*, \forall i \in [k]$, we can rewrite the minimal eigenvalue of the Hessian as follows. For simplicity, we denote $\lambda_{\min}(H) := \lambda_{\min}(\nabla^2 f_{\mathcal{D}}^{\mathrm{ReLU}}(W^*, V^*))$. First we observe that

$$\lambda_{\min}(H) = \min_{\substack{\sum_{i=1}^k \|a_i\|^2 + \|b_i\|^2 = 1 \\ a_{i,1} = 0, \forall i \in [k]}} \mathbb{E}_{x,y} \left[ \left( \sum_{i=1}^k \phi'(u_i^{*\top} x) \phi(v_i^{*\top} y) x^\top a_i + \phi'(v_i^{*\top} y) \phi(u_i^{*\top} x) y^\top b_i \right)^2 \right]. \tag{14}$$

Without loss of generality, we assume $V^* = I$. Set $x = U^* s$, then we have

$$\lambda_{\min}(H) = \min_{\substack{\sum_{i=1}^k \|a_i\|^2 + \|b_i\|^2 = 1 \\ a_{i,1} = 0, \forall i \in [k]}} \mathbb{E}_{x,y} \left[ \left( \sum_{i=1}^k \phi'(s_i) \phi(y_i) s^\top U^{*\top} a_i + \phi'(y_i) \phi(x_i) y^\top b_i \right)^2 \right]$$

$$= \min_{\substack{\sum_{i=1}^k \|a_i\|^2 + \|b_i\|^2 = 1 \\ u^{*(1)} a_i = 0, \forall i \in [k]}} \mathbb{E}_{x,y} \left[ \left( \sum_{i=1}^k \phi'(s_i) \phi(y_i) s^\top a_i + \phi'(y_i) \phi(x_i) y^\top b_i \right)^2 \right],$$

where $u^{*(1)}$ is the first row of $U^*$ and the second equality is because we replace $U^{*\top} a_i$ by $a_i$. In the ReLU case, we have

$$\alpha_{1,0} = \alpha_{1,1} = \alpha_{2,0} = \beta_{1,0} = \beta_{1,1} = \beta_{1,1} = \beta_{2,0} = \beta_{2,2} = \gamma = 1/2.$$

According to Eq. (12), we have

$$\lambda_{\min}(H) \geq \min_{\|A\|_F^2 + \|B\|_F^2 = 1, u^{*(1)} A = 0} C_0 (\|A_o\|_F^2 + \|B_o\|_F^2 + \|A_o + A_o^\top\|_F^2 / 2 + \|B_o + B_o^\top\|_F^2 / 2$$

$$+ \|A_o + B_o^\top\|_F^2 + \|g_A + g_B\|^2$$

$$+ \|A_o 1 + g_A + g_B\|^2 + \|B_o 1 + g_A + g_B\|^2 + (g_A^\top 1 + g_B^\top 1)^2),$$

where $C_0$ is a universal constant. Now we show that there exists a positive number $c_0$ such that $\lambda_{\min}(H) \geq c_0$. If there is no such number, i.e., $\lambda_{\min}(H) = 0$, then we have $A_o = B_o = 0$, $g_A = -g_B$. By the assumption that $u_{1,i}^* \neq 0$ and the condition $u^{*(1)} A = 0$, we have $g_A = g_B = 0$, which violates $\|A\|_F^2 + \|B\|_F^2 = 1$. So $\lambda_{\min}(H) > 0$. An exact value for $c_0$ is postponed to Theorem D.6, which gives the lower bound for the non-orthogonal case. $\qquad \square$

### D.2 Non-orthogonal Case

The restriction of orthogonality on $U, V$ is too strong. We need to consider general non-orthogonal cases. With Gaussian assumption, the non-orthogonal case can be transformed to the orthogonal case according to the following relationship.

**Lemma D.4.** *Let $U \in \mathbb{R}^{d \times k}$ be a full-column rank matrix. Let $g : \mathbb{R}^k \to [0, \infty)$. Define $\lambda(U) = \sigma_1^k(U) / (\prod_{i=1}^k \sigma_i(U))$. Let $\mathcal{D}$ denote the normal distribution. Then*

$$\mathbb{E}_{x \sim \mathcal{D}_d} \left[ g(U^\top x) \right] \geq \frac{1}{\lambda(U)} \mathbb{E}_{z \sim \mathcal{D}_k} \left[ g(\sigma_k(U) z) \right]. \tag{15}$$

**Remark** This lemma transforms $U^\top x$, where the elements of $x$ are mixed, to $\sigma_k(U)z$, where all the elements are independently fed into $g$ with the sacrifices of a condition number of $U$. Using Lemma D.4, we are able to show the PD for non-orthogonal $U^*, V^*$.

*Proof.* Let $P \in \mathbb{R}^{d \times k}$ be the orthonormal basis of $U$, and let $W = [w_1, w_2, \cdots, w_k] = P^\top U \in \mathbb{R}^{k \times k}$.

$$
\mathop{\mathbb{E}}_{x \sim \mathcal{D}_d} [g(U^\top x)]
$$

$$
= \mathop{\mathbb{E}}_{z \sim \mathcal{D}_k} [g(W^\top z)]
$$

$$
= \int (2\pi)^{-k/2} g(W^\top z) e^{-\|z\|^2/2} \mathrm{d}z
$$

$$
= \int (2\pi)^{-k/2} g(s) e^{-\|W^{\dagger\top} s\|^2/2} |\det(W^\dagger)| \mathrm{d}s
$$

$$
\geq \int (2\pi)^{-k/2} g(s) e^{-\sigma_1^2(W^\dagger)\|s\|^2/2} |\det(W^\dagger)| \mathrm{d}s
$$

$$
= \int (2\pi)^{-k/2} g \left( \frac{1}{\sigma_1(W^\dagger)} t \right) e^{-\|t\|^2/2} |\det(W^\dagger)|/\sigma_1^k(W^\dagger) \mathrm{d}t
$$

$$
= \frac{1}{\lambda(W)} \int (2\pi)^{-k/2} g(\sigma_k(W)t) e^{-\|t\|^2/2} \mathrm{d}t
$$

$$
= \frac{1}{\lambda(U)} \mathop{\mathbb{E}}_{z \sim \mathcal{D}_k} [g(\sigma_k(U)z)],
$$

where the third step follows by replacing $z$ by $z = W^{\dagger\top} s$, the fourth step follows by the fact that $\|W^{\dagger\top} s\| \leq \sigma_1(W^\dagger)\|s\|$, and the fifth step follows replacing $s$ by $s = \frac{1}{\sigma_1(W^\dagger)} t$. □

Using Lemma D.4, we are able to provide the lower bound for the minimal eigenvalue for sigmoid and tanh.

**Theorem D.5.** *Assume $\sigma_k(U^*) = \sigma_k(V^*) = 1$. Assume $\beta_{1,1}$ defined in Eq. (10) is 0. Then the minimal eigenvalue of Hessian defined in Eq. (9) can be lower bounded by,*

$$
\lambda_{\min}(H^*) \geq \frac{\rho}{\lambda(U^*)\lambda(V^*)\max\{\kappa(U^*), \kappa(V^*)\}}
$$

*where*

$$
\lambda(U) = \sigma_1^k(U)/(\Pi_{i=1}^k \sigma_i(U)), \kappa(U) = \sigma_1(U)/\sigma_k(U).
$$

*Proof.* Let $P \in \mathbb{R}^{d \times k}, Q \in \mathbb{R}^{d \times k}$ be the orthonormal basis of $U^*, V^*$ respectively. Let $R \in \mathbb{R}^{k \times k}, S \in \mathbb{R}^{k \times k}$ satisfy that $U^* = P \cdot R$ and $V^* = Q \cdot S$. Let $P_\perp \in \mathbb{R}^{d \times (d-k)}, Q_\perp \in \mathbb{R}^{d \times (d-k)}$ be the orthogonal complement of $P, Q$ respectively. Set $a_i = P \cdot s_i + P_\perp \cdot t_i$ and $b_i = Q \cdot p_i + Q_\perp \cdot q_i$. Then we can decompose the minimal eigenvalue problem into three terms.

$$\mathbb{E}_{x,y}\left[\left(\sum_{i=1}^{k}\phi'(u_i^{*\top}x)\phi(v_i^{*\top}y)x^{\top}a_i+\phi'(v_i^{*\top}y)\phi(u_i^{*\top}x)y^{\top}b_i\right)^2\right]$$

$$=\mathbb{E}_{x,y}\left[\left(\sum_{i=1}^{k}\phi'(u_i^{*\top}x)\phi(v_i^{*\top}y)x^{\top}(Ps_i+P_\perp t_i)+\phi'(v_i^{*\top}y)\phi(u_i^{*\top}x)y^{\top}(Qp_i+Q_\perp q_i)\right)^2\right]$$

$$=\underbrace{\mathbb{E}_{x,y}\left[\left(\sum_{i=1}^{k}\phi'(u_i^{*\top}x)\phi(v_i^{*\top}y)x^{\top}Ps_i+\phi'(v_i^{*\top}y)\phi(u_i^{*\top}x)y^{\top}Qp_i\right)^2\right]}_{C_1}$$

$$+\underbrace{\mathbb{E}_{x,y}\left[\left(\sum_{i=1}^{k}\phi'(u_i^{*\top}x)\phi(v_i^{*\top}y)x^{\top}P_\perp t_i\right)^2\right]+\mathbb{E}_{x,y}\left[\left(\sum_{i=1}^{k}\phi'(v_i^{*\top}y)\phi(u_i^{*\top}x)y^{\top}Q_\perp q_i\right)^2\right]}_{C_2},$$

where we omit the terms containing a single independent Gaussian variable, whose expectation is zero. Using Lemma D.4, we can lower bound the term $C_1$ as follows,

$$C_1=\mathbb{E}_{x,y}\left[\left(\sum_{i=1}^{k}\phi'(u_i^{*\top}x)\phi(v_i^{*\top}y)x^{\top}U^*R^{-1}s_i+\phi'(v_i^{*\top}y)\phi(u_i^{*\top}x)y^{\top}V^*S^{-1}p_i\right)^2\right]$$

$$\geq\frac{1}{\lambda(U^*)\lambda(V^*)}\cdot\mathbb{E}_{x,y\sim\mathcal{D}_k}\left[\left(\sum_{i=1}^{k}\phi'(\sigma_k(U^*)x_i))\phi(y_i)x^{\top}R^{-1}s_i\sigma_k(U^*)\right.\right.$$
$$\left.\left.+\phi'(\sigma_k(V^*)y_i)\phi(\sigma_k(U^*)x_i)y^{\top}S^{-1}p_i\sigma_k(V^*))^2\right].$$

And

$$C_2\geq\mathbb{E}_{x,y}\left[\left\|\sum_{i=1}^{k}\phi'(u_i^{*\top}x)\phi(v_i^{*\top}y)t_i\right\|^2\right]$$

$$\geq\frac{1}{\lambda(U^*)\lambda(V^*)}\mathbb{E}_{x,y\sim\mathcal{D}_k}\left[\left\|\sum_{i=1}^{k}\phi'(\sigma_k(U^*)x_i)\phi(\sigma_k(V^*)y_i)t_i\right\|^2\right].$$

Without loss of generality, we assume $\sigma_k(U^*)=\sigma_k(V^*)=1$. Then according to Lemma D.1 and Lemma D.2, we have

$$\lambda_{\min}(H)\geq\frac{1}{\lambda(U^*)\lambda(V^*)\max\{\kappa(U^*),\kappa(V^*)\}}$$
$$\cdot\min\{(\alpha_{2,0}\beta_{2,0}-\alpha_{1,0}^2\beta_{1,0}^2-\beta_{1,0}^2\alpha_{1,1}^2),(\alpha_{2,0}\beta_{2,2}-\alpha_{1,0}^2\beta_{1,2}^2-\gamma^2)\}.$$

Considering the definition of $\rho$ in Eq. (10), we complete the proof. $\qquad\square$

For the ReLU case, we lower bound the minimal eigenvalue of the Hessian for non-orthogonal cases.

**Theorem D.6.** *Consider the activation to be ReLU. Assume $U^*,V^*$ are full-column-rank matrices and $u_{1,i}^*\neq 0,\forall i\in[k]$. Then the minimal eigenvalue of the Hessian of Eq. (13) is lower bounded,*

$$\lambda_{\min}(\nabla^2 f_{\mathcal{D}}^{\text{ReLU}}(W^*,V^*))\gtrsim\frac{1}{\lambda(U^*)\lambda(V^*)}\left(\frac{\min_{i\in[k]}\{|u_{1,i}^*|\}}{(1+\|u^{*(1)}\|)\max\{\|U^*\|,\|V^*\|\}}\right)^2,$$

*where $u^{*(1)}$ is the first row of $U^*$.*

*Proof.* Let $P \in \mathbb{R}^{d \times k}, Q \in \mathbb{R}^{d \times k}$ be the orthonormal basis of $U^*, V^*$ respectively. Let $R \in \mathbb{R}^{k \times k}, S \in \mathbb{R}^{k \times k}$ satisfy that $U^* = P \cdot R$ and $V^* = Q \cdot S$. Let $P_\perp \in \mathbb{R}^{d \times (d-k)}, Q_\perp \in \mathbb{R}^{d \times (d-k)}$ be the orthogonal complement of $P, Q$ respectively. Set $a_i = P \cdot s_i + P_\perp \cdot t_i$ and $b_i = Q \cdot p_i + Q_\perp \cdot q_i$. Similar to the proof of Theorem D.5, Lemma D.2 and Lemma D.3, we have the following.

$$
\mathop{\mathbb{E}}_{x,y} \left[ \left( \sum_{i=1}^k \phi'(u_i^{*\top} x) \phi(v_i^{*\top} y) x^\top a_i + \phi'(v_i^{*\top} y) \phi(u_i^{*\top} x) y^\top b_i \right)^2 \right]
$$

$$
\geq \frac{1}{\lambda(U^*)\lambda(V^*)} \mathop{\mathbb{E}}_{x,y \sim \mathcal{D}_k} \left[ \left( \sum_{i=1}^k \phi'(\sigma_k(U^*)x_i)\phi(y_i)x^\top R^{-1} s_i \sigma_k(U^*) \right. \right.
$$

$$
\left. \left. + \phi'(\sigma_k(V^*)y_i)\phi(\sigma_k(U^*)x_i) y^\top S^{-1} p_i \sigma_k(V^*) \right)^2 \right]
$$

$$
+ \frac{1}{\lambda(U^*)\lambda(V^*)} \mathop{\mathbb{E}}_{x,y \sim \mathcal{D}_k} \left[ \left\| \sum_{i=1}^k \phi'(\sigma_k(U^*)x_i)\phi(\sigma_k(V^*)y_i)t_i \right\|^2 \right]
$$

$$
+ \frac{1}{\lambda(U^*)\lambda(V^*)} \mathop{\mathbb{E}}_{x,y \sim \mathcal{D}_k} \left[ \left\| \sum_{i=1}^k \phi'(\sigma_k(U^*)x_i)\phi(\sigma_k(V^*)y_i)q_i \right\|^2 \right]
$$

$$
\geq \frac{1}{16\lambda(U^*)\lambda(V^*)} (\|\widehat{A}_o\|_F^2 + \|\widehat{B}_o\|_F^2 + \|g_{\widehat{A}} + g_{\widehat{B}}\|^2 + 3(\|\widehat{T}\|_F^2 + \|\widehat{Q}\|_F^2)),
$$

where $\widehat{A} = [R^{-1}s_1, R^{-1}s_2, \cdots, R^{-1}s_k]$, $\widehat{B} = [S^{-1}p_1, S^{-1}p_2, \cdots, S^{-1}p_k]$, $\widehat{T} = [t_1, t_2, \cdots, t_k]$, $\widehat{Q} = [q_1, q_2, \cdots, q_k]$.

Similar to Eq. (14), we can find the minimal eigenvalue of the Hessian by the following constrained minimization problem.

$$
\lambda_{\min}(H) = \min_{\substack{\sum_{i=1}^k \|a_i\|^2 + \|b_i\|^2 = 1 \\ a_{i,1} = 0, \forall i \in [k]}} \mathop{\mathbb{E}}_{x,y} \left[ \left( \sum_{i=1}^k \phi'(u_i^{*\top} x)\phi(v_i^{*\top} y)x^\top a_i + \phi'(v_i^{*\top} y)\phi(u_i^{*\top} x)y^\top b_i \right)^2 \right],
$$

which is lower bounded by the following formula.

$$
\min_{\widehat{A}, \widehat{B}, \widehat{T}, \widehat{P}} \frac{1}{16\lambda(U^*)\lambda(V^*)} (\|\widehat{A}_o\|_F^2 + \|\widehat{B}_o\|_F^2 + \|g_{\widehat{A}} + g_{\widehat{B}}\|^2 + 3(\|\widehat{T}\|_F^2 + \|\widehat{Q}\|_F^2))
$$
$$
\text{s.t. } \|R\widehat{A}\|_F^2 + \|S\widehat{B}\|_F^2 + \|\widehat{T}\|_F^2 + \|\widehat{Q}\|_F^2 = 1 \tag{16}
$$
$$
e_1^\top P R \widehat{A} + e_1^\top P_\perp \widehat{T} = 0
$$

If we assume the minimum of the above formula is $c_1$. We show that $c_1 > 0$ by contradiction. If $c_1 = 0$, then $\widehat{T} = \widehat{Q} = 0$, $\widehat{A}_o = \widehat{B}_o = 0$, $g_{\widehat{A}} = -g_{\widehat{B}}$. Since $\widehat{T} = 0$, we have $e_1^\top P R \widehat{A} = e_1^\top U^* \widehat{A} = 0$. Assuming $(e_1^\top U^*)_i \neq 0, \forall i$, we have $g_{\widehat{A}} = g_{\widehat{B}} = 0$. This violates the condition that $\|R\widehat{A}\|_F^2 + \|S\widehat{B}\|_F^2 + \|\widehat{T}\|_F^2 + \|\widehat{Q}\|_F^2 = 1$.

Now we give a lower bound for $c_1$. First we note,

$$
\|R\widehat{A}\|_F^2 + \|S\widehat{B}\|_F^2 + \|\widehat{T}\|_F^2 + \|\widehat{Q}\|_F^2 \leq \|R\|^2 \|\widehat{A}\|_F^2 + \|S\|^2 \|\widehat{B}\|_F^2 + \|\widehat{T}\|_F^2 + \|\widehat{Q}\|_F^2.
$$

Therefore,

$$
\|\widehat{A}\|_F^2 + \|\widehat{B}\|_F^2 + \|\widehat{T}\|_F^2 + \|\widehat{Q}\|_F^2 \geq \frac{1}{\max\{\|U^*\|^2, \|V^*\|^2\}}.
$$

Also, as $e_1^\top U^* \widehat{A}_o + (e_1^\top U^*) \odot g_{\widehat{A}}^\top + e_1^\top P_\perp \widehat{T} = 0$, where $\odot$ is the element-wise product, we have

$$\|g_{\widehat{A}}\|^2 \le (\frac{1}{\min\{|u_{1,i}^*|\}}(\|u^{*(1)}\|\|\widehat{A}_o\| + \|\widehat{T}\|)^2$$

$$\le \left(\frac{1 + \|u^{*(1)}\|}{\min\{|u_{1,i}^*|\}}\right)^2 2(\|\widehat{A}_o\|_F^2 + \|\widehat{T}\|_F^2).$$

Note that $\|g_{\widehat{A}}\|^2 + \|g_{\widehat{A}} + g_{\widehat{B}}\|^2 \ge \frac{1}{2}\|g_{\widehat{B}}\|^2$. Now let's return to the main part of objective function Eq. (16).

$$\|\widehat{A}_o\|_F^2 + \|\widehat{B}_o\|_F^2 + \|g_{\widehat{A}} + g_{\widehat{B}}\|^2 + 3(\|\widehat{T}\|_F^2 + \|\widehat{Q}\|_F^2)$$

$$\ge \frac{2}{3}(\|\widehat{A}_o\|_F^2 + \|\widehat{T}\|_F^2) + \frac{1}{3}\|\widehat{A}_o\|_F^2 + \|\widehat{B}_o\|_F^2 + \|g_{\widehat{A}} + g_{\widehat{B}}\|^2 + \|\widehat{T}\|_F^2 + \|\widehat{Q}\|_F^2$$

$$\ge \frac{1}{3}\left(\frac{\min\{|u_{1,i}^*|\}}{1 + \|u^{*(1)}\|}\right)^2 \|g_{\widehat{A}}\|^2 + \frac{1}{3}\|\widehat{A}_o\|_F^2 + \|\widehat{B}_o\|_F^2 + \|g_{\widehat{A}} + g_{\widehat{B}}\|^2 + \|\widehat{T}\|_F^2 + \|\widehat{Q}\|_F^2$$

$$\ge \frac{1}{12}\left(\frac{\min\{|u_{1,i}^*|\}}{1 + \|u^{*(1)}\|}\right)^2 (\|g_{\widehat{A}}\|^2 + \|g_{\widehat{B}}\|^2) + \frac{1}{3}\|\widehat{A}_o\|_F^2 + \|\widehat{B}_o\|_F^2 + \|\widehat{T}\|_F^2 + \|\widehat{Q}\|_F^2$$

$$\ge \frac{1}{12}\left(\frac{\min\{|u_{1,i}^*|\}}{1 + \|u^{*(1)}\|}\right)^2 \left(\|g_{\widehat{A}}\|^2 + \|g_{\widehat{B}}\|^2 + \|\widehat{A}_o\|_F^2 + \|\widehat{B}_o\|_F^2 + \|\widehat{T}\|_F^2 + \|\widehat{Q}\|_F^2\right)$$

$$\ge \frac{1}{12}\left(\frac{\min\{|u_{1,i}^*|\}}{(1 + \|u^{*(1)}\|)\max\{\|U^*\|, \|V^*\|\}}\right)^2.$$

Therefore,

$$c_1 \ge \frac{1}{200\lambda(U^*)\lambda(V^*)}\left(\frac{\min\{|u_{1,i}^*|\}}{(1 + \|u^{*(1)}\|)\max\{\|U^*\|, \|V^*\|\}}\right)^2.$$

$\square$

# E  Positive Definiteness of the Empirical Hessian

For any $(U, V)$, the population Hessian can be decomposed into the following $2k \times 2k$ blocks $(i \in [k], j \in [k])$,

$$\frac{\partial^2 f_{\mathcal{D}}(U,V)}{\partial u_i \partial u_j} = \mathop{\mathbb{E}}_{x,y}\left[\phi'(u_i^\top x)\phi'(u_j^\top x)xx^\top \phi(v_i^\top y)\phi(v_j^\top y)\right]$$
$$+ \delta_{ij}\mathop{\mathbb{E}}_{x,y}\left[\left(\phi(U^\top x)^\top \phi(V^\top y) - \phi(U^{*\top} x)^\top \phi(V^{*\top} y)\right)\phi''(u_i^\top x)\phi(v_i^\top y)xx^\top\right]$$

$$\frac{\partial^2 f_{\mathcal{D}}(U,V)}{\partial u_i \partial v_j} = \mathop{\mathbb{E}}_{x,y}\left[\phi'(u_i^\top x)\phi'(v_j^\top y)xy^\top \phi(v_i^\top y)\phi(u_j^\top x)\right]$$
$$+ \delta_{ij}\mathop{\mathbb{E}}_{x,y}\left[\left(\phi(U^\top x)^\top \phi(V^\top y) - \phi(U^{*\top} x)^\top \phi(V^{*\top} y)\right)\phi'(u_i^\top x)\phi'(v_i^\top y)xy^\top\right],$$

(17)

where $\delta_{ij} = 1$ if $i = j$, otherwise $\delta_{ij} = 0$. Similarly we can write the formula for $\frac{\partial^2 f_{\mathcal{D}}(U,V)}{\partial v_i \partial v_j}$ and $\frac{\partial^2 f_{\mathcal{D}}(U,V)}{\partial v_i \partial u_j}$.

Replacing $\mathbb{E}_{x,y}$ by $\frac{1}{|\Omega|}\sum_{(x,y)\in\Omega}$ in the above formula, we can obtain the formula for the corresponding empirical Hessian, $\nabla^2 f_\Omega(U, V)$.

We now bound the difference between $\nabla^2 f_\Omega(U, V)$ and $\nabla^2 f_{\mathcal{D}}(U^*, V^*)$.

**Theorem E.1** (Restatement of Theorem C.2). *For any $\epsilon > 0$, if*

$$n_1 \gtrsim \epsilon^{-2}td\log^2 d, \quad n_2 \gtrsim \epsilon^{-2}t\log d, \quad |\Omega| \gtrsim \epsilon^{-2}td\log^2 d,$$

*then with probability $1 - d^{-t}$, for sigmoid/tanh,*

$$\|\nabla^2 f_\Omega(U, V) - \nabla^2 f_\mathcal{D}(U^*, V^*)\| \lesssim \epsilon + \|U - U^*\| + \|V - V^*\|,$$

*for ReLU,*

$$\|\nabla^2 f_\Omega(U, V) - \nabla^2 f_\mathcal{D}(U^*, V^*)\| \lesssim \left( \left( \frac{\|V - V^*\|}{\sigma_k(V^*)} \right)^{1/2} + \left( \frac{\|U - U^*\|}{\sigma_k(U^*)} \right)^{1/2} + \epsilon \right) (\|U^*\| + \|V^*\|)^2.$$

*Proof.* Define $H(U, V) \in \mathbb{R}^{(2kd) \times (2kd)}$ as a symmetric matrix, whose blocks are represented as

$$
\begin{aligned}
H_{u_i, u_j} &= \underset{x, y}{\mathbb{E}} \left[ \phi'(u_i^\top x) \phi'(u_j^\top x) x x^\top \phi(v_i^\top y) \phi(v_j^\top y) \right], \\
H_{u_i, v_j} &= \underset{x, y}{\mathbb{E}} \left[ \phi'(u_i^\top x) \phi'(v_j^\top y) x y^\top \phi(v_i^\top y) \phi(u_j^\top x) \right].
\end{aligned}
\tag{18}
$$

where $H_{u_i, u_j} \in \mathbb{R}^{d \times d}, H_{u_i, v_j} \in \mathbb{R}^{d \times d}$ correspond to $\frac{\partial^2 f_\mathcal{D}(U, V)}{\partial u_i \partial u_j}, \frac{\partial^2 f_\mathcal{D}(U, V)}{\partial u_i \partial v_j}$ respectively.

We decompose the difference into

$$\|\nabla^2 f_\Omega(U, V) - \nabla^2 f_\mathcal{D}(U^*, V^*)\| \le \|\nabla^2 f_\Omega(U, V) - H(U, V)\| + \|H(U, V) - \nabla^2 f_\mathcal{D}(U^*, V^*)\|.$$

Combining Lemma E.2, E.14, we complete the proof. □

**Lemma E.2.** *For any $\epsilon > 0$, if*

$$n_1 \gtrsim \epsilon^{-2} t d \log^2 d, \quad n_2 \gtrsim \epsilon^{-2} t \log d, \quad |\Omega| \gtrsim \epsilon^{-2} t d \log^2 d,$$

*then with probability $1 - d^{-t}$, for sigmoid/tanh,*

$$\|\nabla^2 f_\Omega(U, V) - H(U, V)\| \lesssim \epsilon + \|U - U^*\| + \|V - V^*\|,$$

*for ReLU,*

$$\|\nabla^2 f_\Omega(U, V) - H(U, V)\| \lesssim \epsilon \|U^*\| \|V^*\|.$$

*Proof.* We can bound $\|\nabla^2 f_\Omega(U, V) - H(U, V)\|$ if we bound each block.

We can show that if

$$n_1 \gtrsim \epsilon^{-2} t d \log^2 d, \quad n_2 \gtrsim \epsilon^{-2} t \log d, \quad |\Omega| \gtrsim \epsilon^{-2} t d \log^2 d,$$

then with probability $1 - d^{-t}$,

$$\left\| \left( \underset{x, y}{\mathbb{E}} - \frac{1}{|\Omega|} \sum_{(x, y) \in \Omega} \right) \left[ \phi'(u_i^\top x) \phi'(u_j^\top x) x x^\top \phi(v_i^\top y) \phi(v_j^\top y) \right] \right\|$$
$$\lesssim \epsilon \|U^*\|^p \|V^*\|^p \qquad \qquad \text{Lemma E.3}$$

$$\left\| \frac{1}{|\Omega|} \sum_{(x, y) \in \Omega} \left[ \left( \phi(U^\top x)^\top \phi(V^\top y) - \phi(U^{*\top} x)^\top \phi(V^{*\top} y) \right) \phi''(u_i^\top x) \phi(v_i^\top y) x x^\top \right] \right\|$$
$$\lesssim \|U - U^*\| + \|V - V^*\| \qquad \qquad \text{Lemma E.6}$$

$$\left\| \left( \underset{x, y}{\mathbb{E}} - \frac{1}{|\Omega|} \sum_{(x, y) \in \Omega} \right) \left[ \phi'(u_i^\top x) \phi'(v_j^\top y) x y^\top \phi(v_i^\top y) \phi(u_j^\top x) \right] \right\|$$
$$\lesssim \epsilon \|U^*\|^p \|V^*\|^p \qquad \qquad \text{Lemma E.7}$$

$$\left\| \frac{1}{|\Omega|} \sum_{(x, y) \in \Omega} \left[ \left( \phi(U^\top x)^\top \phi(V^\top y) - \phi(U^{*\top} x)^\top \phi(V^{*\top} y) \right) \phi'(u_i^\top x) \phi'(v_i^\top y) x y^\top \right] \right\|$$
$$\lesssim \|U - U^*\| + \|V - V^*\|, \qquad \qquad \text{Lemma E.9}$$

where $p = 1$ if $\phi$ is ReLU, $p = 0$ if $\phi$ is sigmoid/tanh.

Note that for ReLU activation, for any given $U, V$, the second term is 0 because $\phi''(z) = 0$ almost everywhere. □

**Lemma E.3.** *If*

$$n_1 \gtrsim \epsilon^{-2} t d \log^2 d, \quad n_2 \gtrsim \epsilon^{-2} t \log d, \quad |\Omega| \gtrsim \epsilon^{-2} t d \log^2 d,$$

*then with probability at least* $1 - d^{-t}$,

$$\left\| \left( \mathbb{E}_{x,y} - \frac{1}{|\Omega|} \sum_{(x,y)\in\Omega} \right) \left[ \phi'(u_i^\top x)\phi'(u_j^\top x)xx^\top \phi(v_i^\top y)\phi(v_j^\top y) \right] \right\| \le \epsilon \|v_i\|^p \|v_j\|^p$$

*where $p = 1$ if $\phi$ is ReLU, $p = 0$ if $\phi$ is sigmoid/tanh.*

*Proof.* Let $B(x,y) = \phi'(u_i^\top x)\phi'(u_j^\top x)xx^\top \phi(v_i^\top y)\phi(v_j^\top y)$. By applying Lemma E.11 and Property $(I) - (III), (VI)$ in Lemma E.4 and Lemma E.5, we have for any $\epsilon > 0$ if

$$n_1 \gtrsim \epsilon^{-2} t d \log^2 d, \quad n_2 \gtrsim \epsilon^{-2} t \log d,$$

then with probability at least $1 - d^{-2t}$,

$$\left\| \mathbb{E}_{x,y}[B(x,y)] - \frac{1}{|S|} \sum_{(x,y)\in S} B(x,y) \right\| \le \epsilon \|v_i\|^p \|v_j\|^p. \tag{19}$$

By applying Lemma E.12 and Property $(I), (III) - (V)$ in Lemma E.4 and Lemma E.5, we have for any $\epsilon > 0$ if

$$n_1 \gtrsim \epsilon^{-1} t d \log^2 d, \quad n_2 \gtrsim \epsilon^{-2} t \log d,$$

then

$$\left\| \frac{1}{n_1} \sum_{l\in[n_1]} (\phi'(u_i^\top x_l)\phi'(u_j^\top x_l))^2 \|x_l\|^2 x_l x_l^\top \right\| \lesssim d,$$

and

$$\left\| \frac{1}{n_2} \sum_{l\in[n_2]} (\phi(v_i^\top y_l)\phi(v_j^\top y_l))^2 \right\| \lesssim \|v_i\|^{2p} \|v_j\|^{2p}.$$

Therefore,

$$\max \left( \left\| \frac{1}{|S|} \sum_{(x,y)\in S} B(x,y)B(x,y)^\top \right\|, \left\| \frac{1}{|S|} \sum_{(x,y)\in S} B(x,y)^\top B(x,y) \right\| \right) \lesssim \epsilon d \|v_i\|^{2p} \|v_j\|^{2p}. \tag{20}$$

We can apply Lemma E.13 and use Eq. (20) and Property (I) in Lemma E.4 and Lemma E.5 to obtain the following result. If

$$|\Omega| \gtrsim \epsilon^{-2} t d \log^2 d,$$

then with probability at least $1 - d^{-2t}$,

$$\left\| \frac{1}{|S|} \sum_{(x,y)\in S} B(x,y) - \frac{1}{|\Omega|} \sum_{(x,y)\in\Omega} B(x,y) \right\| \lesssim \epsilon \|v_i\|^p \|v_j\|^p. \tag{21}$$

Combining Eq. (19) and (21), we finish the proof. $\qquad\square$

**Lemma E.4.** *Define* $T(z) = \phi'(u_i^\top z)\phi'(u_j^\top z)zz^\top$. *If* $z \sim \mathcal{Z}$, $\mathcal{Z} = \mathcal{N}(0, I_d)$ *and* $\phi$ *is ReLU or sigmoid/tanh, the following holds for* $T(z)$ *and any* $t > 1$,

$$(\text{I}) \qquad \Pr_{z \sim \mathcal{Z}} \left[ \|T(z)\| \leq 5td\log n \right] \geq 1 - n^{-1}d^{-t};$$

$$(\text{II}) \qquad \max_{\|a\|=\|b\|=1} \left( \mathbb{E}_{z \sim \mathcal{Z}} \left[ \left( a^\top T(z)b \right)^2 \right] \right)^{1/2} \lesssim 1;$$

$$(\text{III}) \qquad \max \left( \left\| \mathbb{E}_{z \sim \mathcal{Z}}[T(z)T(z)^\top] \right\|, \left\| \mathbb{E}_{z \sim \mathcal{Z}}[T(z)^\top T(z)] \right\| \right) \lesssim d;$$

$$(\text{IV}) \qquad \max_{\|a\|=1} \left( \mathbb{E}_{z \sim \mathcal{Z}} \left[ \left( a^\top T(z)T(z)^\top a \right)^2 \right] \right)^{1/2} \lesssim d;$$

$$(\text{V}) \qquad \left\| \mathbb{E}_{z \sim \mathcal{Z}}[T(z)T(z)^\top T(z)T(z)^\top] \right\| \lesssim d^3;$$

$$(\text{VI}) \qquad \left\| \mathbb{E}_{z \sim \mathcal{Z}}[T(z)] \right\| \lesssim 1.$$

*Proof.* Note that $0 \leq \phi'(z) \leq 1$, therefore (I) can be proved by Proposition 1 of [HKZ12]. (II) − (VI) can be proved by Hölder's inequality. □

**Lemma E.5.** *Define* $T(z) = \phi(v_i^\top z)\phi(v_j^\top z)$. *If* $z \sim \mathcal{Z}$, $\mathcal{Z} = \mathcal{N}(0, I_d)$ *and* $\phi$ *is ReLU or sigmoid/tanh, the following holds for* $T(z)$ *and any* $t > 1$,

$$(\text{I}) \qquad \Pr_{z \sim \mathcal{Z}} \left[ \|T(z)\| \leq 5t\|v_i\|^p\|v_j\|^p \log n \right] \geq 1 - n^{-1}d^{-t};$$

$$(\text{II}) \qquad \max_{\|a\|=\|b\|=1} \left( \mathbb{E}_{z \sim \mathcal{Z}} \left[ \left( a^\top T(z)b \right)^2 \right] \right)^{1/2} \lesssim \|v_i\|^p\|v_j\|^p;$$

$$(\text{III}) \qquad \max \left( \left\| \mathbb{E}_{z \sim \mathcal{Z}}[T(z)T(z)^\top] \right\|, \left\| \mathbb{E}_{z \sim \mathcal{Z}}[T(z)^\top T(z)] \right\| \right) \lesssim \|v_i\|^{2p}\|v_j\|^{2p};$$

$$(\text{IV}) \qquad \max_{\|a\|=1} \left( \mathbb{E}_{z \sim \mathcal{Z}} \left[ \left( a^\top T(z)T(z)^\top a \right)^2 \right] \right)^{1/2} \lesssim \|v_i\|^{2p}\|v_j\|^{2p};$$

$$(\text{V}) \qquad \left\| \mathbb{E}_{z \sim \mathcal{Z}}[T(z)T(z)^\top T(z)T(z)^\top] \right\| \lesssim \|v_i\|^{4p}\|v_j\|^{4p};$$

$$(\text{VI}) \qquad \left\| \mathbb{E}_{z \sim \mathcal{Z}}[T(z)] \right\| \lesssim \|v_j\|^p\|v_i\|^p.$$

*where* $p = 1$ *if* $\phi$ *is ReLU,* $p = 0$ *if* $\phi$ *is sigmoid/tanh.*

*Proof.* Note that $|\phi(z)| \leq |z|^p$, therefore (I) can be proved by Proposition 1 of [HKZ12]. (II)−(VI) can be proved by Hölder's inequality □

**Lemma E.6.** *If*

$$n_1 \gtrsim \epsilon^{-2}td\log^2 d, \quad n_2 \gtrsim \epsilon^{-2}t\log d, \quad |\Omega| \gtrsim \epsilon^{-2}td\log^2 d,$$

*then with probability at least* $1 - d^{-t}$,

$$\left\| \frac{1}{|\Omega|} \sum_{(x,y) \in \Omega} \left[ \left( \phi(U^\top x)^\top \phi(V^\top y) - \phi(U^{*\top} x)^\top \phi(V^{*\top} y) \right) \phi''(u_i^\top x)\phi(v_i^\top y)xx^\top \right] \right\|$$
$$\lesssim (\|U - U^*\| + \|V - V^*\|).$$

*Proof.* We consider the following formula first,

$$\left\| \frac{1}{|\Omega|} \sum_{(x,y) \in \Omega} \left[ \left( (\phi(u_j^\top x) - \phi(u_j^{*\top} x))\phi(v_j^{*\top} y) \right) \phi''(u_i^\top x)\phi(v_i^\top y)xx^\top \right] \right\|$$
$$\leq \left\| \frac{1}{|\Omega|} \sum_{(x,y) \in \Omega} \left[ \left| (u_j - u_j^*)^\top x \right| xx^\top \phi(v_j^{*\top} y)\phi(v_i^\top y) \right] \right\|.$$

Similar to Lemma E.3, we are able to show

$$\left\| \frac{1}{|\Omega|} \sum_{(x,y)\in\Omega} \left[ \left| (u_j - u_j^*)^\top x \right| x x^\top \phi(v_j^{*\top} y)\phi(v_i^\top y) \right] - \mathop{\mathbb{E}}_{(x,y)} \left[ \left| (u_j - u_j^*)^\top x \right| x x^\top \phi(v_j^{*\top} y)\phi(v_i^\top y) \right] \right\|$$
$$\lesssim \|U - U^*\|.$$

Note that by Hölder's inequality, we have,

$$\left\| \mathop{\mathbb{E}}_{(x,y)} \left[ \left| (u_j - u_j^*)^\top x \right| x x^\top \phi(v_j^{*\top} y)\phi(v_i^\top y) \right] \right\| \lesssim \|U - U^*\|.$$

So we complete the proof. $\qquad\square$

**Lemma E.7.** *If*

$$n_1 \gtrsim \epsilon^{-2} t d \log^2 d, \quad n_2 \gtrsim \epsilon^{-2} t \log d, \quad |\Omega| \gtrsim \epsilon^{-2} t d \log^2 d,$$

*then with probability at least $1 - d^{-t}$,*

$$\left\| \left( \mathop{\mathbb{E}}_{x,y} - \frac{1}{|\Omega|} \sum_{(x,y)\in\Omega} \right) \left[ \phi'(u_i^\top x)\phi'(v_j^\top y)xy^\top \phi(v_i^\top y)\phi(u_j^\top x) \right] \right\| \lesssim \epsilon \|v_i\|^p \|u_j\|^p.$$

*Proof.* Let $B(x,y) = M(x)N(y)$, where $M(x) = \phi'(u_i^\top x)\phi(u_j^\top x)x$ and $N(y) = \phi'(v_j^\top y)\phi(v_i^\top y)y^\top$. By applying Lemma E.11 and Property $(I) - (III), (VI)$ in Lemma E.8 , we have for any $\epsilon > 0$ if

$$n_1 \gtrsim \epsilon^{-2} t d \log^2 d, \quad n_2 \gtrsim \epsilon^{-2} t d \log^2 d,$$

then with probability at least $1 - d^{-2t}$,

$$\left\| \mathop{\mathbb{E}}_{x,y} B(x,y) - \frac{1}{|S|} \sum_{(x,y)\in S} B(x,y) \right\| \lesssim \epsilon \|u_j\|^p \|v_i\|^p. \tag{22}$$

By applying Lemma E.12 and Property $(I), (IV) - (VI)$ in Lemma E.8, we have for any $\epsilon > 0$ if

$$n_1 \gtrsim \epsilon^{-2} t d \log^2 d, n_2 \gtrsim \epsilon^{-2} t d \log^2 d,$$

then

$$\left\| \frac{1}{n_1} \sum_{l\in[n_1]} M(x_l)M(x_l)^\top \right\| \lesssim \|u_j\|^{2p}, \quad \left\| \frac{1}{n_2} \sum_{l\in[n_2]} N(y_l)^\top N(y_l) \right\| \lesssim \|v_i\|^{2p}.$$

By applying Lemma E.12 and Property $(I), (IV), (VII), (VIII)$ in Lemma E.8, we have for any $\epsilon > 0$ if

$$n_1 \gtrsim \epsilon^{-2} t d \log^2 d, n_2 \gtrsim \epsilon^{-2} t d \log^2 d,$$

then

$$\left\| \frac{1}{n_1} \sum_{l\in[n_1]} M(x_l)^\top M(x_l) \right\| \lesssim d\|u_j\|^{2p}, \quad \left\| \frac{1}{n_2} \sum_{l\in[n_2]} N(y_l)N(y_l)^\top \right\| \lesssim d\|v_i\|^{2p}.$$

Therefore,

$$\max\left( \left\| \frac{1}{|S|} \sum_{(x,y)\in S} B(x,y)B(x,y)^\top \right\|, \left\| \frac{1}{|S|} \sum_{(x,y)\in S} B(x,y)^\top B(x,y) \right\| \right) \lesssim \epsilon d\|v_i\|^{2p}\|u_j\|^{2p} \tag{23}$$

We can apply Lemma E.13 and Eq. (23) and Property (I) in Lemma E.8 to obtain the following result. If

$$|\Omega| \gtrsim \epsilon^{-2} t d \log^2 d,$$

then with probability at least $1 - d^{-2t}$,

$$\left\| \frac{1}{|S|} \sum_{(x,y) \in S} B(x,y) - \frac{1}{|\Omega|} \sum_{(x,y) \in \Omega} B(x,y) \right\| \leq \epsilon \|v_i\|^p \|u_j\|^p. \qquad (24)$$

Combining Eq. (22) and (24), we finish the proof. $\qquad \square$

**Lemma E.8.** *Define* $T(z) = \phi'(u_i^\top z)\phi(u_j^\top z)z$. *If* $z \sim \mathcal{Z}$, $\mathcal{Z} = \mathcal{N}(0, I_d)$ *and* $\phi$ *is ReLU or sigmoid/tanh, the following holds for* $T(z)$ *and any* $t > 1$,

$$
\begin{array}{ll}
\text{(I)} & \Pr_{z \sim \mathcal{Z}} \left[ \|T(z)\| \leq 5td^{1/2}\|u_j\|^p \log n \right] \geq 1 - n^{-1}d^{-t}; \\[2mm]
\text{(II)} & \left\| \mathbb{E}_{z \sim \mathcal{Z}}[T(z)] \right\| \lesssim \|u_j\|^p; \\[2mm]
\text{(III)} & \max_{\|a\|=\|b\|=1} \left( \mathbb{E}_{z \sim \mathcal{Z}} \left[ \left( a^\top T(z) b \right)^2 \right] \right)^{1/2} \lesssim \|u_j\|^p; \\[2mm]
\text{(IV)} & \max \left\{ \left\| \mathbb{E}_{z \sim \mathcal{Z}}[T(z)T(z)^\top] \right\|, \left\| \mathbb{E}_{z \sim \mathcal{Z}}[T(z)^\top T(z)] \right\| \right\} \lesssim d\|u_j\|^{2p}; \\[2mm]
\text{(V)} & \max_{\|a\|=1} \left( \mathbb{E}_{z \sim \mathcal{Z}} \left[ \left( a^\top T(z)T(z)^\top a \right)^2 \right] \right)^{1/2} \lesssim \|u_j\|^{2p}; \\[2mm]
\text{(VI)} & \left\| \mathbb{E}_{z \sim \mathcal{Z}}[T(z)T(z)^\top T(z)T(z)^\top] \right\| \lesssim d\|u_j\|^{4p}; \\[2mm]
\text{(VII)} & \max_{\|a\|=1} \left( \mathbb{E}_{z \sim \mathcal{Z}} \left[ \left( a^\top T(z)^\top T(z)a \right)^2 \right] \right)^{1/2} \lesssim d\|u_j\|^{2p}; \\[2mm]
\text{(VIII)} & \left\| \mathbb{E}_{z \sim \mathcal{Z}}[T(z)^\top T(z)T(z)^\top T(z)] \right\| \lesssim d^2\|u_j\|^{4p}.
\end{array}
$$

*Proof.* Note that $0 \leq \phi'(z) \leq 1, |\phi(z)| \leq |z|^p$, therefore (I) can be proved by Proposition 1 of [HKZ12]. $(\text{II}) - (\text{VIII})$ can be proved by Hölder's inequality. $\qquad \square$

**Lemma E.9.** *If*

$$n_1 \gtrsim td \log^2 d, \quad n_2 \gtrsim t \log d, \quad |\Omega| \gtrsim td \log^2 d,$$

*then with probability at least* $1 - d^{-t}$,

$$
\left\| \frac{1}{|\Omega|} \sum_{(x,y) \in \Omega} \left[ \left( \phi(U^\top x)^\top \phi(V^\top y) - \phi(U^{*\top} x)^\top \phi(V^{*\top} y) \right) \phi'(u_i^\top x)\phi'(v_i^\top y)xy^\top \right] \right\|
$$
$$
\lesssim \|U - U^*\| + \|V - V^*\|.
$$

*Proof.* We consider the following formula first,

$$
\left\| \frac{1}{|\Omega|} \sum_{(x,y) \in \Omega} \left[ \left( (\phi(u_j^\top x) - \phi(u_j^{*\top} x))\phi(v_j^{*\top} y) \right) \phi'(u_i^\top x)\phi'(v_i^\top y)xy^\top \right] \right\|
$$

Set $M(x) = (\phi(u_j^\top x) - \phi(u_j^{*\top} x))\phi'(u_i^\top x)x$ and $N(y) = \phi(v_j^{*\top} y)\phi'(v_i^\top y)y^\top$ and follow the proof for Lemma E.7. Also note that $\phi$ is Lipschitz, i.e., $|\phi(u_j^\top x) - \phi(u_j^{*\top} x)| \leq |u_j^\top x - u_j^{*\top} x|$. We can show the following. If

$$n_1 \gtrsim td \log^2 d, \quad n_2 \gtrsim t \log d, \quad |\Omega| \gtrsim td \log^2 d,$$

then with probability at least $1 - d^{-t}$,

$$\left\| \left( \frac{1}{|\Omega|} \sum_{(x,y) \in \Omega} - \underset{x,y}{\mathbb{E}} \right) [M(x)N(y)] \right\| \lesssim \|u_j - u_j^*\|.$$

Note that by Hölder's inequality, we have,

$$\| \underset{x,y}{\mathbb{E}} [M(x)N(y)] \| \lesssim \|u_j - u_j^*\|.$$

So we complete the proof.

$\square$

We provide a variation of Lemma B.7 in [ZSJ$^+$17]. Note that the Lemma B.7 [ZSJ$^+$17] requires four properties, we simplify it into three properties.

**Lemma E.10** (Matrix Bernstein for unbounded case (A modified version of bounded case, Theorem 6.1 in [Tro12], A variation of Lemma B.7 in [ZSJ$^+$17]))**.** *Let $\mathcal{B}$ denote a distribution over $\mathbb{R}^{d_1 \times d_2}$. Let $d = d_1 + d_2$. Let $B_1, B_2, \cdots B_n$ be i.i.d. random matrices sampled from $\mathcal{B}$. Let $\overline{B} = \mathbb{E}_{B \sim \mathcal{B}}[B]$ and $\widehat{B} = \frac{1}{n} \sum_{i=1}^n B_i$. For parameters $m \geq 0, \gamma \in (0,1), \nu > 0, L > 0$, if the distribution $\mathcal{B}$ satisfies the following four properties,*

$$\text{(I)} \qquad \underset{B \sim \mathcal{B}}{\Pr}[\|B\| \leq m] \geq 1 - \gamma;$$

$$\text{(II)} \qquad \max \left( \left\| \underset{B \sim \mathcal{B}}{\mathbb{E}}[BB^\top] \right\|, \left\| \underset{B \sim \mathcal{B}}{\mathbb{E}}[B^\top B] \right\| \right) \leq \nu;$$

$$\text{(III)} \qquad \max_{\|a\|=\|b\|=1} \left( \underset{B \sim \mathcal{B}}{\mathbb{E}} \left[ \left( a^\top B b \right)^2 \right] \right)^{1/2} \leq L.$$

*Then we have for any $\epsilon > 0$ and $t \geq 1$, if*

$$n \geq (18t \log d) \cdot ((\epsilon + \|\overline{B}\|)^2 + m\epsilon + \nu)/\epsilon^2 \quad \text{and} \quad \gamma \leq (\epsilon/(2L))^2$$

*with probability at least $1 - d^{-2t} - n\gamma$,*

$$\left\| \frac{1}{n} \sum_{i=1}^n B_i - \underset{B \sim \mathcal{B}}{\mathbb{E}}[B] \right\| \leq \epsilon.$$

*Proof.* Define the event

$$\xi_i = \{\|B_i\| \leq m\}, \forall i \in [n].$$

Define $M_i = \mathbf{1}_{\|B_i\| \leq m} B_i$. Let $\overline{M} = \mathbb{E}_{B \sim \mathcal{B}}[\mathbf{1}_{\|B\| \leq m} B]$ and $\widehat{M} = \frac{1}{n} \sum_{i=1}^n M_i$. By triangle inequality, we have

$$\|\widehat{B} - \overline{B}\| \leq \|\widehat{B} - \widehat{M}\| + \|\widehat{M} - \overline{M}\| + \|\overline{M} - \overline{B}\|. \tag{25}$$

In the next a few paragraphs, we will upper bound the above three terms.

**The first term in Eq.** (25). For each $i$, let $\overline{\xi}_i$ denote the complementary set of $\xi_i$, i.e. $\overline{\xi}_i = [n] \backslash \xi_i$. Thus $\Pr[\overline{\xi}_i] \leq \gamma$. By a union bound over $i \in [n]$, with probability $1 - n\gamma$, $\|B_i\| \leq m$ for all $i \in [n]$. Thus $\widehat{M} = \widehat{B}$.

**The second term in Eq. (25).** For a matrix $B$ sampled from $\mathcal{B}$, we use $\xi$ to denote the event that $\xi = \{\|B\| \le m\}$. Then, we can upper bound $\|\overline{M} - \overline{B}\|$ in the following way,

$$
\begin{aligned}
&\|\overline{M} - \overline{B}\| \\
&= \left\| \mathop{\mathbb{E}}_{B \sim \mathcal{B}}[\mathbf{1}_{\|B\| \le m} \cdot B] - \mathop{\mathbb{E}}_{B \sim \mathcal{B}}[B] \right\| \\
&= \left\| \mathop{\mathbb{E}}_{B \sim \mathcal{B}}\left[ B \cdot \mathbf{1}_{\overline{\xi}} \right] \right\| \\
&= \max_{\|a\|=\|b\|=1} \mathop{\mathbb{E}}_{B \sim \mathcal{B}}\left[ a^\top B b \mathbf{1}_{\overline{\xi}} \right] \\
&\le \max_{\|a\|=\|b\|=1} \mathop{\mathbb{E}}_{B \sim \mathcal{B}}[(a^\top B b)^2]^{1/2} \cdot \mathop{\mathbb{E}}_{B \sim \mathcal{B}}\left[ \mathbf{1}_{\overline{\xi}} \right]^{1/2} && \text{by Hölder's inequality} \\
&\le L \mathop{\mathbb{E}}_{B \sim \mathcal{B}}\left[ \mathbf{1}_{\overline{\xi}} \right]^{1/2} && \text{by Property (IV)} \\
&\le L \gamma^{1/2}, && \text{by } \Pr[\overline{\xi}] \le \gamma \\
&\le \frac{1}{2}\epsilon, && \text{by } \gamma \le (\epsilon/(2L))^2,
\end{aligned}
$$

which is

$$ \|\overline{M} - \overline{B}\| \le \frac{\epsilon}{2}. $$

Therefore, $\|\overline{M}\| \le \|\overline{B}\| + \frac{\epsilon}{2}$.

**The third term in Eq. (25).** We can bound $\|\widehat{M} - \overline{M}\|$ by Matrix Bernstein's inequality [Tro12].

We define $Z_i = M_i - \overline{M}$. Thus we have $\mathop{\mathbb{E}}_{B_i \sim \mathcal{B}}[Z_i] = 0$, $\|Z_i\| \le 2m$, and

$$
\left\| \mathop{\mathbb{E}}_{B_i \sim \mathcal{B}}[Z_i Z_i^\top] \right\| = \left\| \mathop{\mathbb{E}}_{B_i \sim \mathcal{B}}[M_i M_i^\top] - \overline{M} \cdot \overline{M}^\top \right\| \le \nu + \|\overline{M}\|^2 \le \nu + \|\overline{B}\|^2 + \epsilon^2 + \epsilon\|\overline{B}\|.
$$

Similarly, we have $\left\| \mathop{\mathbb{E}}_{B_i \sim \mathcal{B}}[Z_i^\top Z_i] \right\| \le \nu + \|\overline{B}\|^2 + \epsilon^2 + \epsilon\|\overline{B}\|$. Using matrix Bernstein's inequality, for any $\epsilon > 0$,

$$
\Pr_{B_1, \cdots, B_n \sim \mathcal{B}}\left[ \frac{1}{n} \left\| \sum_{i=1}^{n} Z_i \right\| \ge \epsilon \right] \le d \exp\left( -\frac{\epsilon^2 n/2}{\nu + \|\overline{B}\|^2 + \epsilon^2 + \epsilon\|\overline{B}\| + 2m\epsilon/3} \right).
$$

By choosing

$$
n \ge (3t \log d) \cdot \frac{\nu + \|\overline{B}\|^2 + \epsilon^2 + \epsilon\|\overline{B}\| + 2m\epsilon/3}{\epsilon^2/2},
$$

for $t \ge 1$, we have with probability at least $1 - d^{-2t}$,

$$
\left\| \frac{1}{n} \sum_{i=1}^{n} M_i - \overline{M} \right\| \le \frac{\epsilon}{2}.
$$

Putting it all together, we have for $\epsilon > 0$, if

$$
n \ge (18 t \log d) \cdot ((\epsilon + \|\overline{B}\|)^2 + m\epsilon + \nu)/(\epsilon^2) \quad \text{and} \quad \gamma \le (\epsilon/(2L))^2
$$

with probability at least $1 - d^{-2t} - n\gamma$,

$$
\left\| \frac{1}{n} \sum_{i=1}^{n} B_i - \mathop{\mathbb{E}}_{B \sim \mathcal{B}}[B] \right\| \le \epsilon.
$$

$\square$

**Lemma E.11** (Tail Bound for fully-observed rating matrix). *Let $\{x_i\}_{i\in[n_1]}$ be independent samples from distribution $\mathcal{X}$ and $\{y_j\}_{j\in[n_2]}$ be independent samples from distribution $\mathcal{Y}$. Denote $S := \{(x_i, y_j)\}_{i\in[n_1],j\in[n_2]}$ as the collection of all the $(x_i, y_j)$ pairs. Let $B(x,y)$ be a random matrix of $x, y$, which can be represented as the product of two matrices $M(x), N(y)$, i.e., $B(x,y) = M(x)N(y)$. Let $\overline{M} = \mathbb{E}_x M(x)$ and $\overline{N} = \mathbb{E}_y N(y)$. Let $d_x$ be the sum of the two dimensions of $M(x)$ and $d_y$ be the sum of the two dimensions of $N(y)$. Suppose both $M(x)$ and $N(y)$ satisfy the following properties ($z$ is a representative for $x, y$, and $T(z)$ is a representative for $M(x), N(y)$),*

$$(\text{I}) \qquad \Pr_{z\sim\mathcal{Z}}\left[\|T(z)\| \leq m_z\right] \geq 1 - \gamma_z;$$

$$(\text{II}) \qquad \max_{\|a\|=\|b\|=1}\left(\mathbb{E}_{z\sim\mathcal{Z}}\left[\left(a^\top T(z)b\right)^2\right]\right)^{1/2} \leq L_z;$$

$$(\text{III}) \qquad \max\left(\left\|\mathbb{E}_{z\sim\mathcal{Z}}[T(z)T(z)^\top]\right\|, \left\|\mathbb{E}_{z\sim\mathcal{Z}}[T(z)^\top T(z)]\right\|\right) \leq \nu_z.$$

*Then for any $\epsilon_1 > 0, \epsilon_2 > 0$ if*

$$n_1 \geq (18t\log d_x)\cdot(\nu_x + (\|\overline{M}\| + \epsilon_1)^2 + m_x\epsilon_1)/\epsilon_1^2 \quad and \quad \gamma_x \leq (\epsilon_1/(2L_x))^2$$

$$n_2 \geq (18t\log d_y)\cdot(\nu_y + (\epsilon_2 + \|\overline{N}\|)^2 + m_y\epsilon_2)/\epsilon_2^2 \quad and \quad \gamma_y \leq (\epsilon_2/(2L_y))^2$$

*with probability at least $1 - d_x^{-2t} - d_y^{-2t} - n_1\gamma_x - n_2\gamma_y$,*

$$\left\|\mathbb{E}_{x,y}B(x,y) - \frac{1}{|S|}\sum_{(x,y)\in S}B(x,y)\right\| \leq \epsilon_2\|\overline{M}\| + \epsilon_1\|\overline{N}\| + \epsilon_1\epsilon_2. \tag{26}$$

*Proof.* First we note that,

$$\frac{1}{|S|}\sum_{(x,y)\in S}B(x,y) = \frac{1}{n_1 n_2}\left(\sum_{i\in[n_1]}M(x_i)\right)\cdot\left(\sum_{j\in[n_2]}N(y_j)\right),$$

and

$$\mathbb{E}_{x,y}[B(x,y)] = \left(\mathbb{E}_x[M(x)]\right)\left(\mathbb{E}_y[N(y)]\right).$$

Therefore, if we can bound $\|\mathbb{E}_x[M(x)] - \frac{1}{n_1}\sum_{i\in[n_1]}M(x_i)\|$ and the corresponding term for $y$, we are able to prove this lemma.

By the conditions of $M(x)$, the three conditions in Lemma E.10 are satisfied, which completes the proof.

$\square$

**Lemma E.12** (Upper bound for the second-order moment). *Let $\{z_i\}_{i\in[n]}$ be independent samples from distribution $\mathcal{Z}$. Let $T(z)$ be a matrix of $z$. Let $d$ be the sum of the two dimensions of $T(z)$ and $\overline{T} := \mathbb{E}_{z\sim\mathcal{Z}}[T(z)T(z)^\top]$. Suppose $T(z)$ satisfies the following properties.*

$$(\text{I}) \qquad \Pr_{z\sim\mathcal{Z}}\left[\|T(z)\| \leq m_z\right] \geq 1 - \gamma_z;$$

$$(\text{II}) \qquad \max_{\|a\|=1}\left(\mathbb{E}_{z\sim\mathcal{Z}}\left[\left(a^\top T(z)T(z)^\top a\right)^2\right]\right)^{1/2} \leq L_z;$$

$$(\text{III}) \qquad \left\|\mathbb{E}_{z\sim\mathcal{Z}}[T(z)T(z)^\top T(z)T(z)^\top]\right\| \leq \nu_z,$$

*Then for any $t > 1$, if*

$$n \geq (18t\log d)\cdot(\nu_z + (\|\overline{T}\| + \epsilon)^2 + m_z^2)/\epsilon^2 \quad and \quad \gamma_z \leq (\epsilon/(2L_z))^2,$$

*we have with probability at least $1 - d^{-2t} - n\gamma_z$,*

$$\left\| \frac{1}{n} \sum_{i \in [n]} T(z_i)T(z_i)^\top \right\| \leq \left\| \mathbb{E}_{z \sim \mathcal{Z}}[T(z)T(z)^\top] \right\| + \epsilon.$$

*Proof.* The proof directly follows by applying Lemma E.10. $\qquad\square$

**Lemma E.13** (Tail Bound for partially-observed rating matrix). *Given $\{x_i\}_{i \in [n_1]}$ and $\{y_j\}_{j \in [n_2]}$, let's denote $S := \{(x_i, y_j)\}_{i \in [n_1], j \in [n_2]}$ as the collection of all the $(x_i, y_j)$ pairs. Let $\Omega$ also be a collection of $(x_i, y_j)$ pairs, where each pair is sampled from $S$ independently and uniformly. Let $B(x, y)$ be a matrix of $x, y$. Let $d_B$ be the sum of the two dimensions of $B(x, y)$. Define $\overline{B}_S = \frac{1}{|S|} \sum_{(x,y) \in S} B(x, y)$. Assume the following,*

(I) $\quad \|B(x, y)\| \leq m_B, \forall (x, y) \in S,$

(II) $\quad \max \left( \left\| \frac{1}{|S|} \sum_{(x,y) \in S} B(x,y)B(x,y)^\top \right\|, \left\| \frac{1}{|S|} \sum_{(x,y) \in S} B(x,y)^\top B(x,y) \right\| \right) \leq \nu_B.$

*Then we have for any $\epsilon > 0$ and $t \geq 1$, if*

$$|\Omega| \geq (18t \log d_B) \cdot (\nu_B + \|\overline{B}_S\|^2 + m_B\epsilon)/\epsilon^2,$$

*with probability at least $1 - d_B^{-2t}$,*

$$\left\| \overline{B}_S - \frac{1}{|\Omega|} \sum_{(x,y) \in \Omega} B(x, y) \right\| \leq \epsilon.$$

*Proof.* Since each entry in $\Omega$ is sampled from $S$ uniformly and independently, we have

$$\mathbb{E}_\Omega \left[ \frac{1}{|\Omega|} \sum_{(x,y) \in \Omega} B(x, y) \right] = \frac{1}{|S|} \sum_{(x,y) \in S} B(x, y).$$

Applying the matrix Bernstein inequality Theorem 6.1 in [Tro12], we prove this lemma. $\qquad\square$

**Lemma E.14.** *For sigmoid/tanh activation function,*

$$\|H(U, V) - \nabla^2 f_\mathcal{D}(U^*, V^*)\| \lesssim (\|V - V^*\| + \|U - U^*\|),$$

*where $H(U, V)$ is defined as in Eq. (18).*

*For ReLU activation function,*

$$\|H(U, V) - \nabla^2 f_\mathcal{D}(U^*, V^*)\| \lesssim \left( \left( \frac{\|V - V^*\|}{\sigma_k(V^*)} \right)^{1/2} \|U^*\| + \left( \frac{\|U - U^*\|}{\sigma_k(U^*)} \right)^{1/2} \|V^*\| \right) (\|U^*\| + \|V^*\|).$$

*Proof.* We can bound each block, i.e.,

$$\mathbb{E}_{x,y} \left[ \phi'(u_i^\top x)\phi'(u_j^\top x)xx^\top \phi(v_i^\top y)\phi(v_j^\top y) - \phi'(u_i^{*\top} x)\phi'(u_j^{*\top} x)xx^\top \phi(v_i^{*\top} y)\phi(v_j^{*\top} y) \right]. \quad (27)$$

$$\mathbb{E}_{x,y} \left[ \phi'(u_i^\top x)\phi'(v_j^\top y)xy^\top \phi(v_i^\top y)\phi(u_j^\top x) - \phi'(u_i^{*\top} x)\phi'(v_j^{*\top} y)xy^\top \phi(v_i^{*\top} y)\phi(u_j^{*\top} x) \right]. \quad (28)$$

For smooth activations, the bound for Eq. (27) follows by combining Lemma E.15 and Lemma E.16 and the bound for Eq. (28) follows Lemma E.18 and Lemma E.20. For ReLU activation, the bound for Eq. (27) follows by combining Lemma E.15, Lemma E.17 and the bound for Eq. (28) follows Lemma E.18 and Lemma E.19. $\qquad\square$

**Lemma E.15.**

$$\left\| \mathbb{E}_{y \sim \mathcal{D}_d} \left[ (\phi(v_i^\top y) - \phi(v_i^{*\top} y)) \phi(v_j^\top y) \right] \right\| \lesssim \|V^*\|^p \|V - V^*\|.$$

*Proof.* The proof follows the property of the activation function ($\phi(z) \leq |z|^p$) and Hölder's inequality. $\qquad\square$

**Lemma E.16.** *When the activation function is smooth, we have*

$$\left\| \mathbb{E}_{x \sim \mathcal{D}_d} \left[ (\phi'(u_i^\top x) - \phi'(u_i^{*\top} x)) \phi'(u_l^\top x) xx^\top \right] \right\| \lesssim \|U - U^*\|.$$

*Proof.* The proof directly follows Eq. (12) in Lemma D.10 in [ZSJ+17]. $\qquad\square$

**Lemma E.17.** *When the activation function is piece-wise linear with $e$ turning points, we have*

$$\left\| \mathbb{E}_{x \sim \mathcal{D}_d} \left[ (\phi'(u_i^\top x) - \phi'(u_i^{*\top} x)) \phi'(u_l^\top x) xx^\top \right] \right\| \lesssim (e\|U - U^*\| / \sigma_k(U^*))^{1/2}.$$

*Proof.*

$$\left\| \mathbb{E}_{x,y} \left[ (\phi'(u_i^\top x) - \phi'(u_i^{*\top} x)) \phi'(u_l^\top x) xx^\top \right] \right\| \leq \max_{\|a\|=1} \left( \mathbb{E}_{x \sim \mathcal{D}_d} \left[ |\phi'(u_i^\top x) - \phi'(u_i^{*\top} x)| \phi'(u_l^\top x)(x^\top a)^2 \right] \right).$$

Let $P$ be the orthogonal basis of $\mathrm{span}(u_i, u_i^*, u_l)$. Without loss of generality, we assume $u_i, u_i^*, u_l$ are independent, so $P = \mathrm{span}(u_i, u_i^*, u_l)$ is $d$-by-3. Let $[q_i\ q_i^*\ q_l] = P^\top [u_i\ u_i^*\ u_l] \in \mathbb{R}^{3\times3}$. Let $a = Pb + P_\perp c$, where $P_\perp \in \mathbb{R}^{d\times(d-3)}$ is the complementary matrix of $P$.

$$
\begin{aligned}
&\mathbb{E}_{x \sim \mathcal{D}_d} \left[ |\phi'(u_i^\top x) - \phi'(u_i^{*\top} x)| \phi'(u_l^\top x)(x^\top a)^2 \right] \\
=\ &\mathbb{E}_{x \sim \mathcal{D}_d} \left[ |\phi'(u_i^\top x) - \phi'(u_i^{*\top} x)| \phi'(u_l^\top x)(x^\top (Pb + P_\perp c))^2 \right] \\
\lesssim\ &\mathbb{E}_{x \sim \mathcal{D}_d} \left[ |\phi'(u_i^\top x) - \phi'(u_i^{*\top} x)| \phi'(u_l^\top x) \left( (x^\top Pb)^2 + (x^\top P_\perp c)^2 \right) \right] \\
=\ &\mathbb{E}_{x \sim \mathcal{D}_d} \left[ |\phi'(u_i^\top x) - \phi'(u_i^{*\top} x)| \phi'(u_l^\top x)(x^\top Pb)^2 \right] \\
&+ \mathbb{E}_{x \sim \mathcal{D}_d} \left[ |\phi'(u_i^\top x) - \phi'(u_i^{*\top} x)| \phi'(u_l^\top x)(x^\top P_\perp c)^2 \right] \\
=\ &\mathbb{E}_{z \sim \mathcal{D}_3} \left[ |\phi'(q_i^\top z) - \phi'(q_i^{*\top} z)| \phi'(q_l^\top z)(z^\top b)^2 \right] \\
&+ \mathbb{E}_{z \sim \mathcal{D}_3, y \sim \mathcal{D}_{d-3}} \left[ |\phi'(q_i^\top z) - \phi'(q_i^{*\top} z)| \phi'(q_l^\top z)(y^\top c)^2 \right],
\end{aligned}
\tag{29}
$$

where the first step follows by $a = Pb + P_\perp c$, the last step follows by $(a+b)^2 \leq 2a^2 + 2b^2$.

We have $e$ exceptional points which have $\phi''(z) \neq 0$. Let these $e$ points be $p_1, p_2, \cdots, p_e$. Note that if $q_i^\top z$ and $q_i^{*\top} z$ are not separated by any of these exceptional points, i.e., there exists no $j \in [e]$ such that $q_i^\top z \leq p_j \leq q_i^{*\top} z$ or $q_i^{*\top} z \leq p_j \leq q_i^\top z$, then we have $\phi'(q_i^\top z) = \phi'(q_i^{*\top} z)$ since $\phi''(s)$ are zeros except for $\{p_j\}_{j=1,2,\cdots,e}$. So we consider the probability that $q_i^\top z, q_i^{*\top} z$ are separated by any exception point. We use $\xi_j$ to denote the event that $q_i^\top z, q_i^{*\top} z$ are separated by an exceptional point $p_j$. By union bound, $1 - \sum_{j=1}^e \Pr[\xi_j]$ is the probability that $q_i^\top z, q_i^{*\top} z$ are not separated by

any exceptional point. The first term of Equation (29) can be bounded as,

$$\mathop{\mathbb{E}}_{z\sim\mathcal{D}_3}\left[|\phi'(q_i^\top z) - \phi'(q_i^{*\top}z)||\phi'(q_l^\top z)|(z^\top b)^2\right]$$

$$= \mathop{\mathbb{E}}_{z\sim\mathcal{D}_3}\left[\mathbf{1}_{\cup_{j=1}^e \xi_j}|\phi'(q_i^\top z) + \phi'(q_i^{*\top}z)||\phi'(q_l^\top z)|(z^\top b)^2\right]$$

$$\le \left(\mathop{\mathbb{E}}_{z\sim\mathcal{D}_3}\left[\mathbf{1}_{\cup_{j=1}^e \xi_j}\right]\right)^{1/2}\left(\mathop{\mathbb{E}}_{z\sim\mathcal{D}_3}\left[(\phi'(q_i^\top z) + \phi'(q_i^{*\top}z))^2\phi'(q_l^\top z)^2(z^\top b)^4\right]\right)^{1/2}$$

$$\le \left(\sum_{j=1}^e \mathop{\Pr}_{z\sim\mathcal{D}_3}[\xi_j]\right)^{1/2}\left(\mathop{\mathbb{E}}_{z\sim\mathcal{D}_3}\left[(\phi'(q_i^\top z) + \phi'(q_i^{*\top}z))^2\phi'(q_l^\top z)^2(z^\top b)^4\right]\right)^{1/2}$$

$$\lesssim \left(\sum_{j=1}^e \mathop{\Pr}_{z\sim\mathcal{D}_3}[\xi_j]\right)^{1/2}\|b\|^2,$$

where the first step follows by if $q_i^\top z, q_i^{*\top}z$ are not separated by any exceptional point then $\phi'(q_i^\top z) = \phi'(q_i^{*\top}z)$ and the last step follows by Hölder's inequality.

It remains to upper bound $\Pr_{z\sim\mathcal{D}_3}[\xi_j]$. First note that if $q_i^\top z, q_i^{*\top}z$ are separated by an exceptional point, $p_j$, then $|q_i^{*\top}z - p_j| \le |q_i^\top z - q_i^{*\top}z| \le \|q_i - q_i^*\|\|z\|$. Therefore,

$$\mathop{\Pr}_{z\sim\mathcal{D}_3}[\xi_j] \le \mathop{\Pr}_{z\sim\mathcal{D}_3}\left[\frac{|q_i^\top z - p_j|}{\|z\|} \le \|q_i - q_i^*\|\right].$$

Note that $(\frac{q_i^{*\top}z}{\|z\|\|q_i^*\|} + 1)/2$ follows Beta(1,1) distribution which is uniform distribution on $[0,1]$.

$$\mathop{\Pr}_{z\sim\mathcal{D}_3}\left[\frac{|q_i^{*\top}z - p_j|}{\|z\|\|q_i^*\|} \le \frac{\|q_i - q_i^*\|}{\|q_i^*\|}\right] \le \mathop{\Pr}_{z\sim\mathcal{D}_3}\left[\frac{|q_i^{*\top}z|}{\|z\|\|q_i^*\|} \le \frac{\|q_i - q_i^*\|}{\|q_i^*\|}\right] \lesssim \frac{\|q_i - q_i^*\|}{\|q_i^*\|} \lesssim \frac{\|U - U^*\|}{\sigma_k(U^*)},$$

where the first step is because we can view $\frac{q_i^{*\top}z}{\|z\|}$ and $\frac{p_j}{\|z\|}$ as two independent random variables: the former is about the direction of $z$ and the later is related to the magnitude of $z$. Thus, we have

$$\mathop{\mathbb{E}}_{z\in\mathcal{D}_3}[|\phi'(q_i^\top z) - \phi'(q_i^{*\top}z)||\phi'(q_l^\top z)|(z^\top b)^2] \lesssim (e\|U - U^*\|/\sigma_k(U^*))^{1/2}\|b\|^2. \qquad (30)$$

Similarly we have

$$\mathop{\mathbb{E}}_{z\sim\mathcal{D}_3, y\sim\mathcal{D}_{d-3}}\left[|\phi'(q_i^\top z) - \phi'(q_i^{*\top}z)||\phi'(q_l^\top z)|(y^\top c)^2\right] \lesssim (e\|U - U^*\|/\sigma_k(U^*))^{1/2}\|c\|^2. \qquad (31)$$

Finally combining Eq. (30) and Eq. (31) completes the proof.

$\square$

**Lemma E.18.**

$$\left\|\mathop{\mathbb{E}}_{x\sim\mathcal{D}_d}\left[(\phi(u_j^\top x) - \phi(u_j^{*\top}x))\phi'(u_i^\top x)x\right]\right\| \lesssim \|U - U^*\|.$$

*Proof.* First, we can use the Lipschitz continuity of the activation function,

$$\left\|\mathop{\mathbb{E}}_{x\sim\mathcal{D}_d}\left[\phi(u_j^\top x) - \phi(u_j^{*\top}x)\phi'(u_i^\top x)x\right]\right\| \le \max_{\|a\|=1}\left\|\mathop{\mathbb{E}}_{x\sim\mathcal{D}_d}\left[|\phi(u_j^\top x) - \phi(u_j^{*\top}x)|\phi'(u_i^\top x)|x^\top a|\right]\right\|$$

$$\le \max_{\|a\|=1} L_\phi \left\|\mathop{\mathbb{E}}_{x\sim\mathcal{D}_d}\left[|u_j^\top x - u_j^{*\top}x|\phi'(u_i^\top x)|x^\top a|\right]\right\|,$$

where $L_\phi \le 1$ is the Lipschitz constant of $\phi$. Then the proof follows Hölder's inequality. $\square$

**Lemma E.19.** *When the activation function is ReLU,*

$$\left\|\mathop{\mathbb{E}}_{x\sim\mathcal{D}_d}\left[\phi(u_j^{*\top}x)(\phi'(u_i^\top x) - \phi'(u_i^{*\top}x))x\right]\right\| \lesssim (\|U - U^*\|/\sigma_k(U^*))^{1/2}\|u_j\|.$$

*Proof.*

$$\left\| \mathop{\mathbb{E}}_{x\sim\mathcal{D}_d}\left[\phi(u_j^{*\top}x)(\phi'(u_i^{\top}x) - \phi'(u_i^{*\top}x))x\right] \right\| \le \max_{\|a\|=1} \mathop{\mathbb{E}}_{x\sim\mathcal{D}_d}\left[|\phi(u_j^{*\top}x)(\phi'(u_i^{\top}x) - \phi'(u_i^{*\top}x))x^{\top}a|\right].$$

Similar to Lemma E.17, we can show that

$$\max_{\|a\|=1} \mathop{\mathbb{E}}_{x\sim\mathcal{D}_d}\left[|\phi(u_j^{*\top}x)(\phi'(u_i^{\top}x) - \phi'(u_i^{*\top}x))x^{\top}a|\right] \lesssim (\|U - U^*\|/\sigma_k(U^*))^{1/2}\|u_j\|.$$

$\square$

**Lemma E.20.** *When the activation function is sigmoid/tanh,*

$$\left\| \mathop{\mathbb{E}}_{x\sim\mathcal{D}_d}\left[\phi(u_j^{*\top}x)(\phi'(u_i^{\top}x) - \phi'(u_i^{*\top}x))x\right] \right\| \lesssim \|U - U^*\|.$$

*Proof.*

$$\left\| \mathop{\mathbb{E}}_{x\sim\mathcal{D}_d}\left[\phi(u_j^{*\top}x)(\phi'(u_i^{\top}x) - \phi'(u_i^{*\top}x))x\right] \right\|$$
$$\le \max_{\|a\|=1} \mathop{\mathbb{E}}_{x\sim\mathcal{D}_d}\left[|\phi(u_j^{*\top}x)(\phi'(u_i^{\top}x) - \phi'(u_i^{*\top}x))x^{\top}a|\right]$$
$$\lesssim \max_{\|a\|=1} \mathop{\mathbb{E}}_{x\sim\mathcal{D}_d}\left[|(u_i^{\top}x - u_i^{*\top}x)x^{\top}a|\right]$$
$$\lesssim \|U - U^*\|.$$

$\square$

### E.1 Local Linear Convergence

Given Theorem 3.2, we are now able to show local linear convergence of gradient descent for sigmoid and tanh activation function.

**Theorem E.21** (Restatement of Theorem 3.3). *Let $[U^c, V^c]$ be the parameters in the $c$-th iteration. Assuming $\|U^c - U^*\| + \|V^c - V^*\| \lesssim 1/(\lambda^2\kappa)$, then given a fresh sample set, $\Omega$, that is independent of $[U^c, V^c]$ and satisfies the conditions in Theorem 3.2, the next iterate using one step of gradient descent, i.e., $[U^{c+1}, V^{c+1}] = [U^c, V^c] - \eta\nabla f_\Omega(U^c, V^c)$, satisfies*

$$\|U^{c+1} - U^*\|_F^2 + \|V^{c+1} - V^*\|_F^2 \le (1 - M_l/M_u)(\|U^c - U^*\|_F^2 + \|V^c - V^*\|_F^2)$$

*with probability $1 - d^{-t}$, where $\eta = \Theta(1/M_u)$ is the step size and $M_l \gtrsim 1/(\lambda^2\kappa)$ is the lower bound on the eigenvalues of the Hessian and $M_u \lesssim 1$ is the upper bound on the eigenvalues of the Hessian.*

*Proof.* In order to show the linear convergence of gradient descent, we first show that the Hessian along the line between $[U^c, V^c]$ and $[U^*, V^*]$ are positive definite w.h.p..

The idea is essentially building a $d^{-1/2}\lambda^{-2}\kappa^{-1}$-net for the line between the current iterate and the optimal. In particular, we set $d^{1/2}$ points $\{[U^a, V^a]\}_{a=1,2,\cdots,d^{1/2}}$ that are equally distributed between $[U^c, V^c]$ and $[U^*, V^*]$. Therefore, $\|U^{a+1} - U^a\| + \|V^{a+1} - V^a\| \lesssim d^{-1/2}\lambda^{-2}\kappa^{-1}$

Using Lemma E.22, we can show that for any $[U, V]$, if there exists a value of $a$ such that $\|U - U^a\| + \|V - V^a\| \lesssim d^{-1/2}\lambda^{-2}\kappa^{-1}$, then

$$\|\nabla^2 f_\Omega(U, V) - \nabla^2 f_\Omega(U^a, V^a)\| \lesssim \lambda^{-2}\kappa^{-1}.$$

Therefore, for every point $[U, V]$ in the line between $[U^c, V^c]$ and $[U^*, V^*]$, we can find a fixed point in $\{[U^a, V^a]\}_{a=1,2,\cdots,d^{1/2}}$, such that $\|U - U^a\| + \|V - V^a\| \lesssim d^{-1/2}\lambda^{-2}\kappa^{-1}$. Now applying union bound for all $a$, we have that w.p. $1 - d^{-t}$, for every point $[U, V]$ in the line between $[U^c, V^c]$ and $[U^*, V^*]$,

$$M_l I \preceq \nabla^2 f_\Omega(U, V) \preceq M_u,$$

where $M_l = \Omega(\lambda^{-2}\kappa^{-1})$ and $M_u = O(1)$. Note that the upper bound of the Hessian is due to the fact that $\phi$ and $\phi'$ are bounded.

Given the positive definiteness of the Hessian along the line between the current iterate and the optimal, we are ready to show the linear convergence. First we set the stepsize for the gradient descent update as $\eta = 1/M_u$ and use notation $W := [U, V]$ to simplify the writing.

$$\|W^{c+1} - W^*\|_F^2$$
$$= \|W^c - \eta\nabla f_\Omega(W^c) - W^*\|_F^2$$
$$= \|W^c - W^*\|_F^2 - 2\eta\langle\nabla f_\Omega(W^c), (W^c - W^*)\rangle + \eta^2\|\nabla f_\Omega(W^c)\|_F^2$$

Note that

$$\nabla f_\Omega(W^c) = \left(\int_0^1 \nabla^2 f_\Omega(W^* + \xi(W^c - W^*))d\xi\right)\text{vec}(W^c - W^*).$$

Define $H \in \mathbb{R}^{(2kd)\times(2kd)}$,

$$H = \left(\int_0^1 \nabla^2 f_\Omega(W^* + \xi(W^c - W^*))d\xi\right).$$

By the result provided above, we have

$$M_l I \preceq H \preceq M_u I. \tag{32}$$

Now we upper bound the norm of the gradient,

$$\|\nabla f_\Omega(W^c)\|_F^2 = \langle H\text{vec}(W^c - W^*), H\text{vec}(W^c - W^*)\rangle \leq M_u\langle\text{vec}(W^c - W^*), H\text{vec}(W^c - W^*)\rangle.$$

Therefore,

$$\|W^{c+1} - W^*\|_F^2$$
$$\leq \|W^c - W^*\|_F^2 - (-\eta^2 M_u + 2\eta)\langle\text{vec}(W^c - W^*), H\text{vec}(W^c - W^*)\rangle$$
$$\leq \|W^c - W^*\|_F^2 - (-\eta^2 M_u + 2\eta)M_l\|W^c - W^*\|_F^2$$
$$= \|W^c - W^*\|_F^2 - \frac{M_l}{M_u}\|W^c - W^*\|_F^2$$
$$\leq (1 - \frac{M_u}{M_l})\|W^c - W^*\|_F^2$$

$\square$

**Lemma E.22.** *Let the activation function be tan/sigmoid. For given $U^a, V^a$ and $r > 0$, if*

$$n_1 \gtrsim \epsilon^{-2}td\log^2 d, \quad n_2 \gtrsim \epsilon^{-2}t\log d, \quad |\Omega| \gtrsim \epsilon^{-2}td\log^2 d,$$

*then with probability $1 - d^{-t}$,*

$$\sup_{\|U-U^a\|+\|V-V^a\|\leq r}\|\nabla^2 f_\Omega(U, V) - \nabla^2 f_\Omega(U^a, V^a)\| \lesssim d^{1/2}\cdot r$$

*Proof.* We consider each block of the Hessian as defined in Eq (17). In particular, we show that if

$$n_1 \gtrsim \epsilon^{-2}td\log^2 d, \quad n_2 \gtrsim \epsilon^{-2}t\log d, \quad |\Omega| \gtrsim \epsilon^{-2}td\log^2 d,$$

then with probability $1 - d^{-t}$,

$$\left\| \frac{1}{|\Omega|} \sum_{(x,y)\in\Omega} [(\phi'(u_i^\top x)\phi'(u_j^\top x)\phi(v_i^\top y)\phi(v_j^\top y) - \phi'(u_i^{a\top} x)\phi'(u_j^{a\top} x)\phi(v_i^{a\top} y)\phi(v_j^{a\top} y))xx^\top] \right\|$$

$$\lesssim (\|u_i - u_i^a\| + \|u_j - u_j^a\| + \|v_i - v_i^a\| + \|v_j - v_j^a\|)d^{1/2};$$

by Lemma E.23

$$\left\| \frac{1}{|\Omega|} \sum_{(x,y)\in\Omega} \left[ \left(\phi(U^\top x)^\top \phi(V^\top y) - \phi(U^{*\top} x)^\top \phi(V^{*\top} y)\right) \phi''(u_i^\top x)\phi(v_i^\top y)xx^\top \right.\right.$$

$$\left.\left. - \left(\phi(U^{a\top} x)^\top \phi(V^{a\top} y) - \phi(U^{*\top} x)^\top \phi(V^{*\top} y)\right) \phi''(u_i^{a\top} x)\phi(v_i^{a\top} y)xx^\top \right] \right\|$$

$$\lesssim (\|U - U^a\| + \|V - V^a\|)d^{1/2}$$

by Lemma E.24

$$\left\| \frac{1}{|\Omega|} \sum_{(x,y)\in\Omega} [(\phi'(u_i^\top x)\phi'(v_j^\top y)\phi(v_i^\top y)\phi(u_j^\top x) - \phi'(u_i^{a\top} x)\phi'(v_j^{a\top} y)\phi(v_i^{a\top} y)\phi(u_j^{a\top} x)) \, xy^\top] \right\|$$

$$\lesssim (\|u_i - u_i^a\| + \|u_j - u_j^a\| + \|v_i - v_i^a\| + \|v_j - v_j^a\|)d^{1/2}$$

by Lemma E.25

$$\left\| \frac{1}{|\Omega|} \sum_{(x,y)\in\Omega} \left[ \left(\phi(U^\top x)^\top \phi(V^\top y) - \phi(U^{*\top} x)^\top \phi(V^{*\top} y)\right) \phi'(u_i^\top x)\phi'(v_i^\top y)xy^\top \right.\right.$$

$$\left.\left. - \left(\phi(U^{a\top} x)^\top \phi(V^{a\top} y) - \phi(U^{*\top} x)^\top \phi(V^{*\top} y)\right) \phi'(u_i^{a\top} x)\phi'(v_i^{a\top} y)xy^\top \right] \right\|$$

$$\lesssim (\|U - U^a\| + \|V - V^a\|)d^{1/2}$$

by Lemma E.26

$\square$

**Lemma E.23.** *If*

$$n_1 \gtrsim \epsilon^{-2}td\log^2 d, \quad n_2 \gtrsim \epsilon^{-2}t\log d, \quad |\Omega| \gtrsim \epsilon^{-2}td\log^2 d,$$

*then with probability at least $1 - d^{-t}$,*

$$\left\| \frac{1}{|\Omega|} \sum_{(x,y)\in\Omega} [(\phi'(u_i^\top x)\phi'(u_j^\top x)\phi(v_i^\top y)\phi(v_j^\top y) - \phi'(u_i^{a\top} x)\phi'(u_j^{a\top} x)\phi(v_i^{a\top} y)\phi(v_j^{a\top} y))xx^\top] \right\|$$

$$\lesssim (\|u_i - u_i^a\| + \|u_j - u_j^a\| + \|v_i - v_i^a\| + \|v_j - v_j^a\|)d^{1/2}$$

*Proof.* Note that

$$\phi'(u_i^\top x)\phi'(u_j^\top x)\phi(v_i^\top y)\phi(v_j^\top y) - \phi'(u_i^{a\top} x)\phi'(u_j^{a\top} x)\phi(v_i^{a\top} y)\phi(v_j^{a\top} y)$$

$$= \phi'(u_i^\top x)\phi'(u_j^\top x)\phi(v_i^\top y)\phi(v_j^\top y) - \phi'(u_i^{a\top} x)\phi'(u_j^\top x)\phi(v_i^\top y)\phi(v_j^\top y)$$

$$+ \phi'(u_i^{a\top} x)\phi'(u_j^\top x)\phi(v_i^\top y)\phi(v_j^\top y) - \phi'(u_i^{a\top} x)\phi'(u_j^{a\top} x)\phi(v_i^\top y)\phi(v_j^\top y)$$

$$+ \phi'(u_i^{a\top} x)\phi'(u_j^{a\top} x)\phi(v_i^\top y)\phi(v_j^\top y) - \phi'(u_i^{a\top} x)\phi'(u_j^{a\top} x)\phi(v_i^{a\top} y)\phi(v_j^\top y)$$

$$+ \phi'(u_i^{a\top} x)\phi'(u_j^{a\top} x)\phi(v_i^{a\top} y)\phi(v_j^\top y) - \phi'(u_i^{a\top} x)\phi'(u_j^{a\top} x)\phi(v_i^{a\top} y)\phi(v_j^{a\top} y) \quad (33)$$

Let's consider the first term in the above formula. The other terms are similar.

$$\left\| \frac{1}{|\Omega|} \sum_{(x,y)\in\Omega} [(\phi'(u_i^\top x) - \phi'(u_i^{a\top} x))\phi'(u_j^\top x)\phi(v_i^\top y)\phi(v_j^\top y)xx^\top] \right\|$$

$$\leq \left\| \frac{1}{|\Omega|} \sum_{(x,y)\in\Omega} [\|u_i - u_i^a\|\|x\|xx^\top] \right\|$$

which is because both $\phi'(\cdot)$ and $\phi(\cdot)$ are bounded and Lipschitz continuous. Applying the unbounded matrix Bernstein Inequality Lemma E.10, we can bound

$$\left\| \frac{1}{|\Omega|} \sum_{(x,y)\in\Omega} \left[ \|u_i - u_i^a\| \|x\| xx^\top \right] \right\| \lesssim \|u_i - u_i^a\| d^{1/2}$$

Since both $\phi'(\cdot)$ and $\phi(\cdot)$ are bounded and Lipschitz continuous, we can easily extend the above inequality to other cases and finish the proof. $\square$

**Lemma E.24.** *If*

$$n_1 \gtrsim \epsilon^{-2} t d \log^2 d, \quad n_2 \gtrsim \epsilon^{-2} t \log d, \quad |\Omega| \gtrsim \epsilon^{-2} t d \log^2 d,$$

*then with probability at least $1 - d^{-t}$,*

$$\left\| \frac{1}{|\Omega|} \sum_{(x,y)\in\Omega} \left[ \left( \phi(U^\top x)^\top \phi(V^\top y) - \phi(U^{*\top} x)^\top \phi(V^{*\top} y) \right) \phi''(u_i^\top x)\phi(v_i^\top y) xx^\top \right. \right.$$
$$\left. \left. - \left( \phi(U^{a\top} x)^\top \phi(V^{a\top} y) - \phi(U^{*\top} x)^\top \phi(V^{*\top} y) \right) \phi''(u_i^{a\top} x)\phi(v_i^{a\top} y) xx^\top \right] \right\|$$
$$\lesssim (\|U - U^a\| + \|V - V^a\|) d^{1/2}$$

*Proof.* Since for sigmoid/tanh, $\phi, \phi', \phi''$ are all Lipschitz continuous and bounded, the proof of this lemma resembles the proof for Lemma E.23. $\square$

**Lemma E.25.** *If*

$$n_1 \gtrsim \epsilon^{-2} t d \log^2 d, \quad n_2 \gtrsim \epsilon^{-2} t \log d, \quad |\Omega| \gtrsim \epsilon^{-2} t d \log^2 d,$$

*then with probability at least $1 - d^{-t}$,*

$$\left\| \frac{1}{|\Omega|} \sum_{(x,y)\in\Omega} \left[ \left( \phi'(u_i^\top x)\phi'(v_j^\top y)\phi(v_i^\top y)\phi(u_j^\top x) - \phi'(u_i^{a\top} x)\phi'(v_j^{a\top} y)\phi(v_i^{a\top} y)\phi(u_j^{a\top} x) \right) xy^\top \right] \right\|$$
$$\lesssim (\|u_i - u_i^a\| + \|u_j - u_j^a\| + \|v_i - v_i^a\| + \|v_j - v_j^a\|) d^{1/2}$$

*Proof.* Do the similar splits as Eq. (33) and let's consider the following case,

$$\left\| \frac{1}{|\Omega|} \sum_{(x,y)\in\Omega} \left[ \left( \phi'(u_i^\top x) - \phi'(u_i^{a\top} x) \right) \phi'(v_j^\top y)\phi(v_i^\top y)\phi(u_j^\top x) xy^\top \right] \right\|.$$

Setting $M(x) = \left( \phi'(u_i^\top x) - \phi'(u_i^{a\top} x) \right) \phi(u_j^\top x) x$, $N(y) = \phi'(v_j^\top y)\phi(v_i^\top y) y^\top$ and using the fact that $\|\phi'(u_i^\top x) - \phi'(u_i^{a\top} x)\| \le \|u_i - u_i^a\| \|x\|$, we can follow the proof of Lemma E.7 to show if

$$n_1 \gtrsim \epsilon^{-2} t d \log^2 d, \quad n_2 \gtrsim \epsilon^{-2} t \log d, \quad |\Omega| \gtrsim \epsilon^{-2} t d \log^2 d,$$

then with probability at least $1 - d^{-t}$,

$$\left\| \frac{1}{|\Omega|} \sum_{(x,y)\in\Omega} \left[ \left( \phi'(u_i^\top x) - \phi'(u_i^{a\top} x) \right) \phi'(v_j^\top y)\phi(v_i^\top y)\phi(u_j^\top x) xy^\top \right] \right\| \le \|u_i - u_i^a\| d^{1/2}$$

$\square$

**Lemma E.26.** *If*

$$n_1 \gtrsim \epsilon^{-2} t d \log^2 d, \quad n_2 \gtrsim \epsilon^{-2} t \log d, \quad |\Omega| \gtrsim \epsilon^{-2} t d \log^2 d,$$

*then with probability at least $1 - d^{-t}$,*

$$\left\| \frac{1}{|\Omega|} \sum_{(x,y)\in\Omega} \left[ \left( \phi(U^\top x)^\top \phi(V^\top y) - \phi(U^{*\top} x)^\top \phi(V^{*\top} y) \right) \phi'(u_i^\top x)\phi'(v_i^\top y) xy^\top \right. \right.$$
$$\left. \left. - \left( \phi(U^{a\top} x)^\top \phi(V^{a\top} y) - \phi(U^{*\top} x)^\top \phi(V^{*\top} y) \right) \phi'(u_i^{a\top} x)\phi'(v_i^{a\top} y) xy^\top \right] \right\|$$
$$\lesssim (\|U - U^a\| + \|V - V^a\|) d^{1/2}$$

*Proof.* Since for sigmoid/tanh, $\phi, \phi', \phi''$ are all Lipschitz continuous and bounded, the proof of this lemma resembles the proof for Lemma E.25. □