[Reviews · NeurIPS 2019]

Reviewer 1



This paper considers the problem of Non-linear Inductive Matrix Completion (NIMC) in a deep learning formulation. In NIMC one is given a query set, an item set and a few query-item relevance values, and the goal is to learn the query-item relevance function. The main contribution of the paper is to provide theoretical guarantees for using a one hidden layer network to estimate that function via an L2 loss. In particular, this can be thought of as two one-layer networks, one learning the embedding of the queries and the other the embedding of the items, while the relevance function is taken to be the inner product of the outputs of the two networks. The authors prove that for sigmoid and tanh activation functions the objective function is locally strongly convex around the global optimum and that stochastic gradient descent converges linearly if initialized sufficiently well. Interestingly, the authors show that even though ReLu is harder to analyze, locally strong convexity is also possible if one suitably removes some redundancy from the objective function. The paper is mostly well written (especially the first half of it), with well-informed literature, featuring significant and novel results. Technical correctness was also checked to the best of the reviewer’s ability. However, there are some weaknesses that the authors should address: 1. Even though the analysis contains sufficiently novel elements, it is quite similar in structure with some of the papers already appearing in the references. I believe it would be fruitful for the community if the authors actually acknowledge this and in fact describe very carefully what is standard methodology and what is novel. In that spirit, there is a lot of room for improvement in section 3.3. 2. Along the same lines, i find the writing in section 3.4 unsatisfactory. It is too cryptic and it is not essential for a short paper: this type of initialization has already been discussed in references appearing in the paper and a short mention that this can be applied should be enough. Hence i suggest reducing section 3.4 to a short sentence. 3. Instead, more experiments are needed: a) please show the convergence rates for sigmoid and tanh. b) please compare results using “good” and random initialization and c) please show some more realistic experiments with higher values of d,k,n and real data. 4. Please clarify: removing the redundancy from the objective function requires some knowledge about the optimal solution, correct? If so, what is the applicability of this result?

Reviewer 2



In my opinion the paper is very well written. I particularly like that they explain their theoretical results in intuitive words. This paper provides theoretical recovery guarantees for non-linear inductive matrix completion (NIMC) for several activation functions such as sigmoid and ReLu. They show that this non-convex problem behaves as convex near the optimum, and give initialization conditions to "fall" near the optimum. As the authors explain, NIMC tightly related to a one-layer neural network. Providing theoretical guarantees for this type of problems is indeed a very fundamental one, and arguably one of the most important in our field this days. However, the real problem of interest lies in deep networks, as opposed to the single-layer case. In fact, single layers are rather easier to analyze, and have in fact been analyzed quite a bit (e.g., their own references: Zhong et. al, Recovery guarantees for one-hidden-layer neural networks, ICML 2017; Li and Yuan, Convergence Analysis of Two-layer Neural Networks with ReLU Activation, NeurIPS 2017, and more). Hence, while the paper does provide new theory for NIMC, I believe the authors are overselling the importance/novelty of their contribution.

Reviewer 3



The paper considers the recommendation systems, where each user/item is represented by d-dim vector x/y. The goal is to predict the user-iter, relevance score. The score is modeled as dot(Phi(Ux) , Phi(Vy)). This can be considered as a single layer network with parameters U and V and activation function Phi. The paper analyses the squared loss objective function in this system for sigmoid and ReLu activations. The paper studies local geometry of the loss function and shows that, close to the optima, the function is strongly convex. - My main concern is the practical application of this approach. In table 1, the RMSE results of NIMC and IMC are shown on movielens datasets. These RMSEs are very large for these datasets. Simple SVD methods are able to get RMSE error of .95 in ml100k. So the first question is why both NIMC and IMC work so bad in these two datasets? - Another concern is the generalization capability of the recommendation systems. It is well known in the literature that most methods suffer from overfitting since the rating matrix is sparse. So most methods stop training long before getting close to the local/global solutions, which are able to reasonably reconstruct the matrix perfectly. The paper shows that for a simple one-layer neural network, close to the optima, the function is strongly convex. But, considering the facts about generalization, do the authors think convexity around the optimal solution is helpful in understanding why neural network based methods perform better than the linear ones?

[Author Response · NeurIPS 2019]

We thank all the reviewers for their constructive feedback!

**To Reviewer #1.**   1. We agree (and will acknowledge more explicitly) that the overall proof program is similar to
existing results in the area. However, NIMC problem presents two key challenges: a) The Hessian has entangled terms
for items' features and queries' features, which are challenging to handle. 2) In addition to non-convexity arising due to
non-linearity of the activation function which standard 1-2 hidden layer NNs also face, we have to handle additional
noise/uncertainty due to missing ratings, and provide strong sample complexity bounds for the results to be meaningful.

2. We'll reduce section 3.4 to a short sentence.

3. Here we provide more experimental results in Fig. 1. We use sigmoid as the activation function, and set $k = 10, d = $
$100$, which are larger than those in the paper ($k = 5, d = 10$). We set the initialization as $W^{(0)} = (1 - \alpha)W^* + \alpha W^{(r)}$,
where $W^*$ is the ground truth, $W^{(r)}$ is a Gaussian random matrix, and $\alpha \in [0, 1]$. In (a), $\alpha = 0.1, n = 1000$, and
$m = 10000$. In (b), $n = 500$. In (c), $\alpha = 0.1$. The other settings are same as those in the paper. As we can see,
(a) shows how the objective value converges, which is almost linear. (b) shows that when the initialization is purely
random ($\alpha = 1$), gradient descent doesn't converge to the ground truth. In the paper, when $k = 5, d = 10$, pure random
initialization still converges to the ground truth. We believe that it is because when $k, d$ are larger, random initialization
can be further away from the ground truth. Hence, gradient descent can get stuck in local optima more easily. Finally,
comparing (c) with Fig. 1(a) of the paper, we can obtain a similar conclusion, i.e., when n is sufficiently large, the
number of observed ratings required for successful recovery remains the same.

(a) log(obj) v.s. iteration.          (b) Recovery rate v.s. initializations.          (c) Recovery rate v.s. $(m, n)$.

4. To remove the redundancy in the ReLU case, we assume that $u_{1,i}^*$ is nonzero for all $i \in [k]$ and know the number of
positives in $\{u_{1,i}^*\}_{i=1,\cdots,k}$. Note that if the columns of $U$ and the columns of $V$ do the same permutation, the output
doesn't change. Without loss of generality, we can assume $\{u_{1,i}^*\}_{i=1,\cdots,k_+}$ ($0 \le k_+ \le k$) are positive and and the
remaining $\{u_{1,i}^*\}_{i=k_++1,\cdots,k}$ are negative. So if we fix $u_{1,i} = 1$ for all $i \le k_+$ and $u_{1,i} = -1$ for all $i > k_+$, we can
remove the redundancy and the target solutions for $U$ and $V$ are $u_{:,i} = u_{:,i}^*/|u_{1,i}^*|$ and $v_{:,i} = v_{:,i}^*|u_{1,i}^*|$ respectively.

**To Reviewer #2.**    We will like to stress that Non-linear Inductive Matrix Completion is a significantly different
architecture than the 1-layer NNs and hence theoretical analyses for the two are quite different. As mentioned in
response to R1, while the high level approach is same, we have to deal with non-linearity of NNs along with the noise
due to missing ratings and entangled Hessian due to non-linearity in both item's features and query's features. These
challenges require a significantly different analysis than existing results.

**To Reviewer #3.**    Movielens dataset: our main goal in these experiments is to study the problem in the *inductive*
setting, i.e., to predict ratings for *new* users. R3's observations for collaborative filtering (CF) are valid but they apply
only to *transductive* setting which does not allow for new users.

a) SVD based solution: SVD based CF does not predict ratings for new users and hence does not apply in the inductive
setting. Furthermore, as we are predicting ratings for completely new users, for which only weak features are available,
naturally the resulting RMSE is worse than the results for the standard collaborative filtering settings (where several
ratings of a user are available a priori).

b) Generalization error: as mentioned above, the only information about new users is their relatively weak features,
hence non-linear methods can extract more information from them compared to the linear ones, and might be the reason
for the superior performance of NIMC over IMC.

[Meta-Review · NeurIPS 2019]

This paper analyzes nonlinear matrix completion. It gives some guarantees on using 1-layer neural nets as features for matrix completion. Reviewers found the paper has some interesting insights.